# Necrotic debris and STING exert therapeutically relevant effects on tumor cholesterol homeostasis

Sampath Katakam[1], Santosh Anand[2] ®, Patricia Martin[1], Nicolo Riggi[1] ®, Ivan Stamenkovic[1] ®

**Malignant tumors commonly display necrosis, which invariably triggers an inflammatory response that supports tumor growth. However, the effect on tumor cells of necrotic debris, or damage-associated molecular patterns (DAMPs) released by dying cells is unknown. Here, we addressed the effect of DAMPs on primary Ewing sarcoma (EwS) cells and cell lines grown in 3D (spheroids) and 2D culture. We show that DAMPs promote the growth of EwS spheroids but not 2D cultures and that the underlying mechanism implicates an increase in cholesterol load in spheroids. In contrast, stimulation of the nucleic acid sensor signaling platform STING by its ligand cyclic GMP-AMP decreases the tumor cell cholesterol load and reduces their tumor initiating ability. Overexpression of STING or stimulation with cyclic GMP-AMP opposes the growth stimulatory effect of DAMPs and synergizes with the cholesterol synthesis inhibitor simvastatin to inhibit tumor growth. Our observations show that modulation of cholesterol homeostasis is a major effect of necrotic cell debris and STING and suggest that combining STING agonists with statins may help control tumor growth.**

## Introduction

Necrosis is a common feature of most solid malignant tumors and a major consequence of chemotherapy. The debris released by necrotic cells include a wide range of damage/danger-associated molecular patterns (DAMPs) that stimulate cell surface and intracellular pattern recognition receptors (PRRs), which transduce signals to trigger a pro-inflammatory response, primarily in macrophages and mast cells (1, 2). PRRs include the TLR family, NOD-like receptors (NLRs), and a variety of additional intracellular nucleic acid sensors (1) that recognize exogenous or endogenous cytosolic DNA. A key DNA sensor in the cytosol is cyclic guanosine monophosphate (GMP)-AMP synthase (cGAS), which catalyzes the synthesis of the second messenger cyclic GMP-AMP (cGAMP) (3, 4, 5, 6). cGAMP binds to the DNA sensing adaptor stimulator of interferon

genes (STING), which plays an important role in the innate immune response by controlling type I and type III interferon expression (7).

Inflammation provides a multitude of diverse mediators that can promote tumor growth and dissemination, and its maintenance, not only by necrotic debris but also by a variety of tumor cell- and tumor microenvironment (TME)-derived factors, provides a mechanism for continued support of tumor progression (8, 9). Recent work has shown that by expressing phosphatidylserine on their surface, apoptotic cell debris can stimulate macrophage activation and cytokine production, thereby contributing to tumor growth (10). Their growth promoting effect could be inhibited by resolvins (10). Whereas, similar to apoptotic bodies, necrotic cell-derived DAMPs may stimulate tumor growth indirectly by contributing to the generation of an inflammatory TME, they may also affect tumor cell behavior directly, as tumor cells express a variable repertoire of PRRs. However, the effect of DAMPs on tumor cells is relatively unexplored.

We addressed the effect of necrotic cell debris on the growth of primary Ewing sarcoma (EwS) cells and cell lines in vitro as well as that of the corresponding xenotransplants in vivo. Ewing sarcoma is the second most common bone malignancy in children and young adults, characterized by a marked tendency to relapse after therapy and high metastatic proclivity. Although aggressive multimodal therapy has significantly improved survival of patients with localized disease at diagnosis (about 70% survival at 5 yr), the tumors frequently relapse and metastatic lesions dramatically darken the prognosis, reducing the 5-yr survival to about 25% (11, 12). Ewing sarcoma emerges after one of several possible reciprocal chromosomal translocations in permissive cells, which leads to the formation of a fusion gene encoding a protein that functions as an aberrant transcription factor. The most common chromosomal translocation, occurring in 85–90% of tumors, is t(22;11)(q24;q12), which fuses the 5′ sequence of *EWSR1* to 3′ sequences of the gene encoding the ets family transcription factor FLI-1 (13). There is strong evidence to suggest that the resulting fusion protein bears dominant if not sole responsibility for EwS pathogenesis (12). The structure of the EWS-FLI-1 protein, however, is not amenable to targeting by small molecule inhibitors (14), leaving open the quest for alternative therapeutic strategies. Ewing sarcoma often displays

[1]Experimental Pathology Service, Centre Hospitalier Universitaire Vaudois and Department of Research and Education, University of Lausanne, Lausanne, Switzerland [2]Department of Informatics, Systems, and Communications (DISCo), University of Milano-Bicocca, Milan, Italy

Correspondence: Ivan.Stamenkovic@unil.ch

extensive necrosis even without treatment and typically undergoes massive necrosis after chemotherapy. However, the high frequency of relapse reflects therapeutic resistance of subpopulations of EwS cells whose growth and possibly dissemination may be aided by cell debris and the associated tissue remodeling. Ewing sarcoma may therefore provide a suitable model to address the effect of necrotic debris on tumor cells.

Using 3D (spheroid) and 2D models of primary EwS and EwS cell lines, we observed necrotic cell supernatants containing DAMPs stimulate growth of EwS spheroids but not 2D cultures by augmenting tumor cell uptake and synthesis of cholesterol. In an effort to assess the effects of stimulating a range of individual DAMP sensors by their ligands, we observed that engagement of STING by its natural ligand cGAMP opposes the effect of DAMPs and inhibits EwS cell proliferation in vitro and the corresponding tumor growth in vivo. In contrast to DAMPs, activation of STING by cGAMP decreased cholesterol synthesis and augmented cholesterol efflux from EwS cells. The debris released by necrotic cells, including nucleic acids, may therefore influence tumor growth by modulating cholesterol homeostasis. Accordingly, we examined the effect of the statin family of cholesterol synthesis inhibitors on 3D EwS growth in conjunction with DAMPs, cGAMP, and modulation of STING expression and observed that statins inhibit the growth stimulatory effect of DAMPs and strongly synergize with the growth inhibitory effect of STING. Our observations suggest that modulation of cholesterol metabolism in EwS cells has an important effect on their proliferation and tumor initiating ability and that statins in combination with STING agonists may provide a potentially attractive strategy to curtail EwS growth.

## Results

### DAMPs stimulate 3D EwS growth

Crude suspensions of necrotic cell debris (henceforth referred to as DAMPs) and apoptotic bodies were prepared from EwS cells after three to five cycles of freeze–thawing (15) and treatment with camptothecin for 24 h (16), respectively. Cells undergoing necrosis stained positive with propidium iodide (PI) and negative for annexin V (Fig S1A). After centrifugation, the pellets from these preparations were discarded and the supernatants collected and used for subsequent experiments. Camptothecin treatment-derived cell fragments were sorted into annexin V–negative and annexin V–positive (apoptotic bodies) fractions (Fig S1A) and the positive fractions were used in subsequent experiments. Primary EwS 3D cultures (P1 and P2) that were established earlier (17, 18), were also grown as monolayers and two EwS cell lines (A673 and TC252, labeled C1 and C2, respectively) were grown in 3D as well as in monolayer culture. DAMPs (final dilution 1:20) or apoptotic bodies (final dilution 1:10) were added to and maintained in cell cultures for 24 h and the cells assessed for changes in proliferation in vitro (Fig 1A). Whereas exposure of primary as well as A673 (C1) EwS cell 3D cultures to DAMPs resulted in their twofold increased growth, addition of DAMP suspensions to the same cells grown as adherent monolayers did not affect their proliferation significantly (Fig 1B

and C). Treatment of primary 3D cultures with apoptotic bodies had no marked effect on growth in vitro (Figs 1B and S1B).

To assess the effect of DAMPs on tumor growth in vivo, 10,000 3D culture-derived EwS cells were injected beneath the kidney capsule of NOD-SCID-common-gamma (NSG) mice with or without DAMPs (1:10 DAMP:cell suspension volume ratio) or apoptotic bodies (1:5 apoptotic body: cell suspension ratio, Fig 1A) and tumor growth was assessed over a 5-wk period. Whereas both DAMPs and apoptotic bodies significantly enhanced 3D culture–derived tumor growth in vivo ($P < 0.05$), DAMPs did not affect 2D culture–derived tumor growth (Fig 1D and E). The mechanism by which apoptotic bodies stimulate tumor growth in vivo has recently been shown to involve activation of myeloid cells, primarily macrophages, within the TME, by apoptotic body-associated phosphatidylserine (10). However, the mechanism by which DAMPs augment 3D-cultured tumor cell and the corresponding tumor growth in vivo has yet to be elucidated.

### Stimulation by DAMPs causes changes in the expression of genes associated with cholesterol metabolism of EwS cells

To begin to address the possible mechanism underlying the observed DAMP-mediated increase in EwS spheroid growth, we conducted RNA-seq analysis of 3D- and 2D-cultured P1 cells stimulated with DAMPs for 4 h (Fig 2A). Control cells were cultured under identical conditions but with substitution of the DAMP suspension by an equal amount of solvent. Using a twofold cutoff for the change in gene expression, we identified 549 and 621 up-regulated and 759 and 273 down-regulated genes in spheroids and 2D-cultured cells, respectively.

The most strongly enriched Gene Ontology (GO) terms included inflammatory response, metabolic process, biological process and signal transduction (Fig S2A). Whereas representation of most of the more selective terms within these four global categories differed only quantitatively between 3D- and 2D-cultured cells, several pathways within the term metabolic process were represented in 3D- but not in 2D-cultures, including lipid biosynthetic process, positive regulation of catabolic process, regulation of lipid transport and steroid biosynthetic process (Fig 2B). Gene Set Enrichment Analysis (GSEA) of unstimulated 3D and 2D cultured cells showed marked enrichment of the terms reactome cholesterol biosynthesis and hallmark cholesterol homeostasis in spheroids (Fig S2B). The top 10 DAMP-induced transcripts, based on 3D-cultured cell responses (Fig S2C), revealed a cholesterol acetyltransferase gene (SOAT), implicated in converting cholesterol to less toxic cholesterol esters, to be highly induced in 3D but not in 2D cultures. Interrogation of the KEGG pathway database revealed that spheroid and 2D-cultured cells displayed marked differences in term overrepresentation with relatively limited overlap (of the total of 549 and 621 up-regulated genes in spheroids and 2D-cultured cells, respectively, only 81 were shared, Fig 2C). Remarkably, enrichment of the term cholesterol was observed in the 3D-culture response only. Comparison of 3D- and 2D culture responses focusing on inflammation and cholesterol metabolism, using a 1.5-fold cutoff for gene expression change, revealed genes associated with inflammation to be strongly enriched in 2D cultures (Fig 2D) and most of the up-regulated genes that overlapped the 2D and 3D-cultures to be related to the inflammatory response. In contrast,

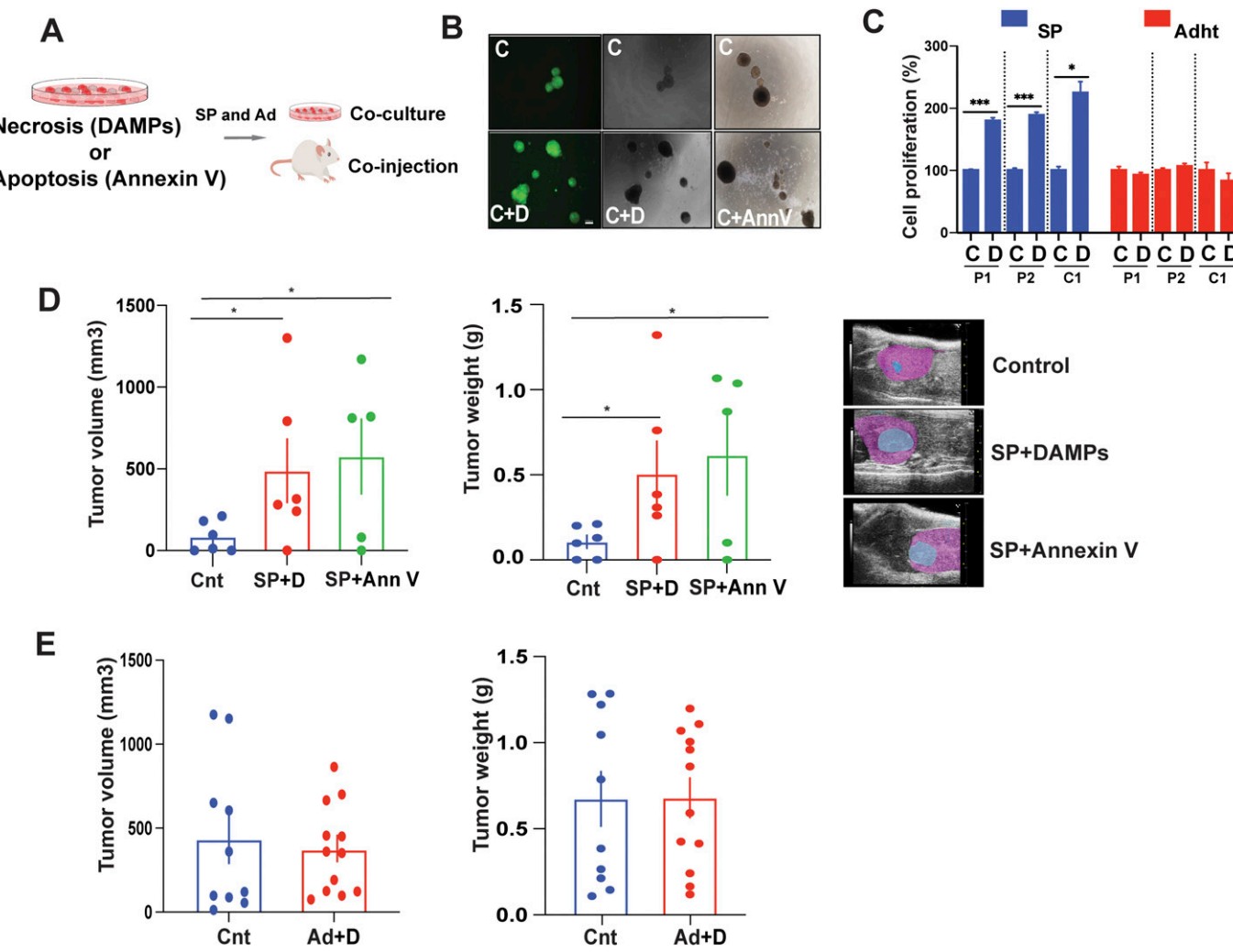

**Figure 1. Damage-associated molecular patterns (DAMPs) stimulate 3D EwS growth.**
**(A)** Experimental setup: Supernatants containing debris (DAMPs) from freeze–thaw–induced EwS cell death and sorted annexin V–positive apoptotic bodies from camptothecin-treated EwS cells were co-cultured in vitro with EwS spheroids and adherent cells and the cells assessed for proliferation. EwS spheroids and adherent cells were also co-injected with DAMP and apoptotic body suspensions into NSG mice. **(B)** GFP-labeled spheroids (P1) were incubated with DAMPs, annexin V–positive sorted cell debris or in culture medium only for 48 h, and images were taken by light and fluorescent microscopy. Scale bar = 100 $\mu$m. Representative experiment images shown for three independent experiments. (C, control; D, DAMPs; AnnV, annexin V). **(C)** Spheroids and adherent cells were incubated with DAMPs for 5 d, and proliferation was assessed by Cell Titer-Glo assay, normalized to untreated controls. Data indicate mean values ± SEM for three independent experiments in replicate. (C, Control; D, DAMPs). **(D)** Tumorigenicity of spheroids alone and spheroids co-injected with DAMPs and with annexin V–positive cell bodies (n = 5–6 mice/group). Tumor volumes were measured weekly by ultrasonography. The average tumor volume (mm³, left panels) and tumor weight (gm, right panels) ± SEM are shown (SP, spheres; Cnt, control; D, DAMPs; AnnV, annexin V). Ultrasonogram images (right) 6 wk after injection of cells beneath the kidney capsule (blue, tumor tissue; pink, kidney tissue). **(E)** In vivo tumorigenicity of co-injected adherent cells with DAMPs and DAMPs alone. n = 10–12 tumors/group. Average tumor volumes were measured every 5 d by electronic calipers. Tumor volume (mm³), (E); left panels: tumor weight (g) in (E); right panels: Ad, adherent; Cnt, control; D, DAMPs; **(C, D, E)** *P < 0.05, **P < 0.01, and ***P < 0.001 by two-tailed unpaired t test. See also Fig S1.

cholesterol metabolism-associated transcripts were significantly enriched only in 3D cultures (Fig 2D).

Based on these observations, we explored the changes in cholesterol metabolism induced by DAMPs and their possible effect on the proliferation of 3D cultured EwS cells. Examination of selected transcripts associated with cholesterol homeostasis revealed higher expression of genes implicated in cholesterol synthesis, esterification and uptake in 3D- than in 2D-cultured cells and a correspondingly more robust up-regulation of their expression in response to DAMPs (Fig 2E). In contrast, genes associated with cholesterol efflux were more highly expressed in 2D-cultures but their up-

regulation upon stimulation with DAMPs appeared to display a similar trend in 3D and 2D cultures (Fig 2E). At least five gene sets implicated in defined aspects of lipid metabolism displayed predominantly increased expression in 3D-cultured P1 cells in response to DAMPs (Fig 2F). GSEA plots revealed marked enrichment of the terms cholesterol biosynthetic process, steroid biosynthetic process and cholesterol metabolic process in 3D cultures exposed to DAMPs (Fig 2G).

To validate the observed transcriptomic changes in response to DAMPs, we selected a panel of genes associated with inflammation and lipid metabolism and assessed the change in their expression in P1, P2 and C1 spheroids and adherent cells by qRT-PCR. Genes

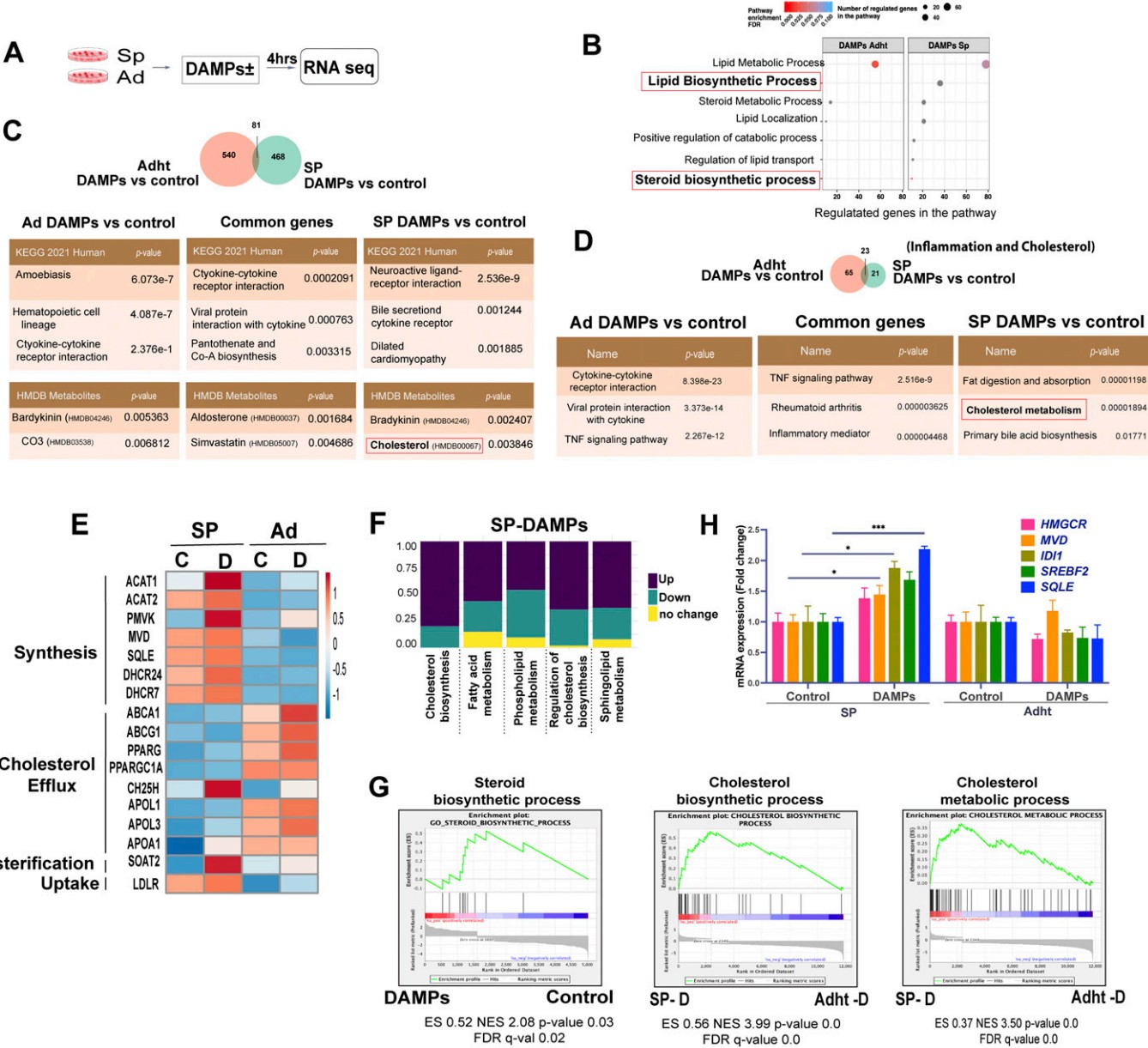

**Figure 2. Stimulation of EwS cells by damage-associated molecular patterns (DAMPs) causes changes in the expression of genes associated with cholesterol metabolism.**
**(A)** Experimental setup for RNA seq analysis of spheroids and adherent cells treated with DAMPs (4 h). **(B)** Enrichment of selected Gene Ontology terms associated with metabolic process pathways in DAMP versus control-treated spheroids and adherent cells based on RNA-seq data. **(C)** Venn diagram showing the overlap between up-regulated genes by DAMP treatment of P1 spheroids and adherent cells (>2-fold up-regulation compared with solvent-treated cells, Top). Enrichment of KEGG Pathway terms and HMDB Metabolites using the Enricher platform (bottom). **(D)** Venn diagram of the overlap between up-regulated inflammation-associated (Hallmark Inflammatory response–Gene Set Enrichment Analysis) and cholesterol metabolism target genes (four lipid metabolism-related gene sets from the Reactome database) in P1 spheroids and adherent cells treated with DAMPs (>1.5-fold up-regulation compared with solvent-treated cells; Top). **(E)** Heat map showing the effect of DAMPs on selected genes that regulate cholesterol homeostasis from RNA-seq data in P1 spheroids (SP) and adherent (Adht) cells. (C, Control; D, DAMPs). **(F)** Stacked bar chart illustrating the fractions of up/down-regulated and unchanged gene expression for five categories of lipid metabolism–related gene sets in the RNA seq data of DAMP-treated spheroids. **(G)** Gene Set Enrichment Analysis plots for the enrichment of steroid biosynthetic process, cholesterol biosynthetic process, and cholesterol metabolic process signatures in the above RNA-seq data. **(H)** qPCR analysis of selected mevalonate pathway synthesis genes in P1 spheroids and adherent cells after treatment with DAMPs or solvent (control) for 24 h. Data indicate mean values ± SEM. Each experiment was performed at least three times. *$P < 0.05$, **$P < 0.01$, and ***$P < 0.001$ by two-tailed unpaired $t$ test. See also Fig S2.

associated with inflammation included CXCL10, CCL5, IFN-$\beta$1, and IFN$\lambda$1, whereas genes associated with the mevalonate pathway, which is responsible for the synthesis of cholesterol and other isoprenoids (19), included HMGCoA reductase (HMGCR), mevalonate diphosphate decarboxylase (MVD), isopentenyl diphosphate isomerase (IDI1), sterol regulatory element-binding transcription factor 2 (SREBF2), and squalene epoxidase (SQLE). DAMPs induced the expression of the genes implicated in inflammation in both 2D- and 3D-cultures (Fig S2D), whereas they up-regulated the mevalonate pathway transcripts in 3D-cultured EwS cells only (Figs 2H and S2E).

### Changes in cholesterol metabolism drive DAMP-dependent proliferation of EwS cell spheroids

To address possible differences in the metabolic rate of 3D- and 2D-cultured cells, we assessed their spare respiratory capacity (SRC), defined as the difference between basal and maximal respiratory capacity, which reflects the amount of extra ATP that can be generated by oxidative phosphorylation in the event of a sudden increase in energy demand. The SRC was calculated by measuring the oxygen consumption rate of the cells, using a Seahorse instrument, after sequential exposure to oligomycin, carbonyl cyanide-4-(trifluoromethoxy)phenylhydrazone (FCCP), and rotenone/antimycin A. By inhibiting ATP synthase, oligomycin rapidly hyperpolarizes the mitochondrial membrane, thereby preventing protons from passing through the respiratory chain complexes. FCCP, which uncouples oxidative phosphorylation, reverses the hyperpolarized state caused by oligomycin by carrying protons across the mitochondrial inner membrane. Finally, rotenone/antimycin inhibit mitochondrial complexes I and III, completely halting mitochondrial respiration. Whereas resting state proliferation of 2D and 3D cells varied according to the cells examined (C1 cells proliferated more rapidly in 2D than in 3D culture but P2 cell spheroids proliferated faster than their monolayer counterparts), the SRC was three to fourfold greater in spheroids than in 2D cultures, irrespective of cell identity (Fig 3A). EwS spheroids may therefore dispose of more adequate means than monolayer cultures to adapt to stress.

To determine whether the increased expression of genes associated with inflammation and cholesterol metabolism might play a role in the observed DAMP-induced proliferation of 3D cultured EwS cells, spheroids of both primary EwS cultures and A673 cells were stimulated with several of the up-regulated mediators of inflammation, including CXCL10, CCL5, IL1-$\beta$, and CCL2. However, only minor changes in proliferation were observed, with variable responses among the cells tested (Fig S3A) suggesting that none of the tested mediators alone could explain the observed DAMP-induced proliferation. Based on the observation that DAMPs did not stimulate proliferation of adherent cells despite inducing the same and additional mediators of inflammation compared to spheroids, we did not pursue the investigation of the role of inflammatory mediators in DAMP-mediated promotion of 3D EwS cell growth.

Instead, we interrogated the effect of DAMPs on cholesterol uptake and synthesis based on the observations that numerous genes implicated in these two functions were up-regulated in EwS cells grown in 3D but not in 2D culture (Fig 3B). We first addressed possible changes in cellular cholesterol uptake and content after 3–5 d of stimulation with DAMPs using filipin as an indicator of cholesterol load. Filipin has been widely used for cytochemical localization of cholesterol in biological membranes, based on its ability to bind to unesterified 3$\beta$-hydroxy-sterols forming 25 nm complexes, but it is also applied to mark free cholesterol in cultured cells. Interaction with cholesterol alters the filipin absorption and fluorescence spectra, allowing visualization with a fluorescence microscope (20). 3D-cultured EwS cells displayed a two to fourfold greater baseline cholesterol load, as assessed by filipin fluorescence intensity and cholesterol quantification, than their 2D-cultured counterparts (Fig 3C and D). DAMPs further increased the intracellular cholesterol of 3D-cultured C1 cells, as measured by filipin fluorescence (Fig 3E–G), and cholesterol quantification revealed an up to threefold increase in cholesterol load (Fig 3H). In contrast, DAMPs either had no effect or even decreased the cholesterol load of 2D-cultured cells (Fig 3H). The baseline and the DAMP-induced total cholesterol load in spheroids were reduced by statins (Fig 3I) and by the cholesterol-depleting agent methyl-$\beta$-cyclodextrin (MBCD) (Fig S3D). DAMPs increased cholesterol uptake in EwS spheroids but not in the corresponding adherent cells (Fig 3J).

To determine whether cholesterol has a growth stimulatory effect on EwS, spheroids from primary EwS and EwS cell lines were cultured in the presence of cholesterol or mevalonic acid (1 mM each) for 5 d and assessed for proliferation. Cholesterol and mevalonic acid stimulated EwS 3D cell growth and the effect was partially neutralized by 50 $\mu$M simvastatin (Fig 3K) and MBCD (Fig S3B–D). Together, these observations suggest that the increase in cholesterol load, resulting from augmentation in both uptake and synthesis, constitutes at least one mechanism by which DAMPs promote EwS cell growth in 3D culture.

### Stimulation of STING by cGAMP decreases EwS cell growth in vitro and in vivo

The DAMP preps used to stimulate EwS cells consisted of a crude suspension of debris released by cells undergoing necrosis. To attempt to identify candidate DAMPs or categories thereof that participate in the observed promotion of spheroid growth we interrogated the possible role of nucleic acids. DNAse, RNAse, and benzonase treatment of crude DAMP suspensions (Fig S4A) had either no effect on DAMP-mediated enhancement of 3D-EwS cell growth or increased it slightly, particularly when the samples were treated with RNAse (Fig S4B). These observations suggest that the nucleic acids present in the crude DAMP preps cannot explain the observed increase in 3D-culture growth and that they may in fact exert an attenuating effect on growth.

To determine whether any of the known PRRs might be implicated in DAMP-mediated stimulation of 3D EwS culture growth, we addressed the effect of stimulating a panel of individual DAMP sensors by their respective ligands (Fig 4A). We first determined the expression repertoire of PRRs in primary EwS 3D cultures and found that EwS cells express TLRs 1–10 as well as the inflammasome NLRP3, the receptor for advanced glycation products (AGER), the cytoplasmic nucleic acid recognition receptor DDX58/RIG1 and STING, as assessed by RT-PCR (Fig S4C). We then stimulated a panel

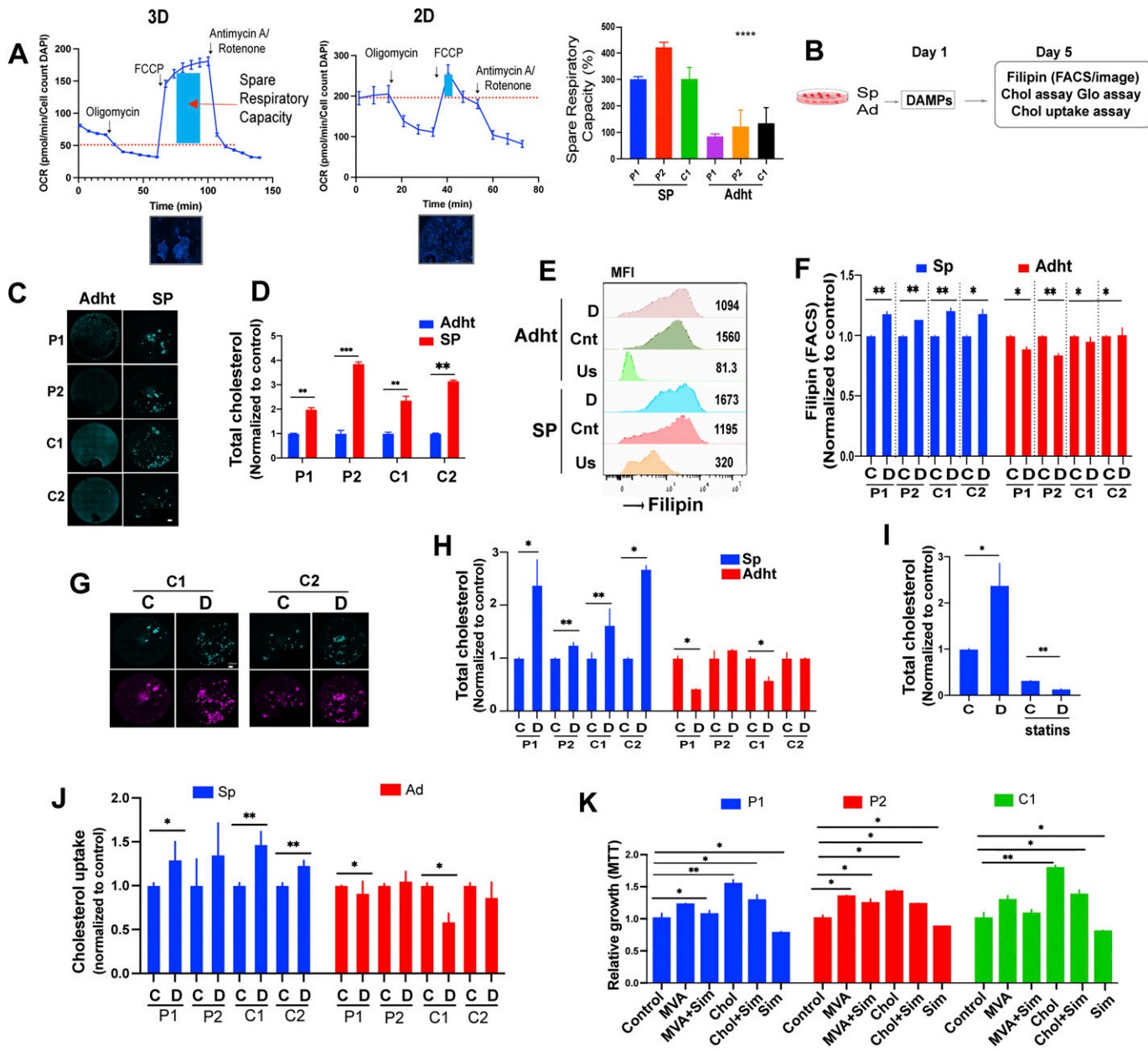

**Figure 3. EwS spheroids have greater spare respiratory capacity (SRC) than 2D-cultured cells and changes in cholesterol metabolism drive damage-associated molecular pattern (DAMP)–dependent proliferation of EwS spheroids.**

**(A)** SRC of P2 spheroids (3D) and adherent (2D) cells along with DAPI staining (4× magnification). Scale bar 1,000 $\mu$m. Right panel: bar graph showing quantification of the SRC (%), as calculated from the oxygen consumption rate of the indicated spheroids (SP) and adherent (Adht) cells in a Seahorse instrument ($n \pm 15$ from representative values of two independent experiments). Oxygen consumption rate values were normalized for fluorescently identified cell counts (DAPI). Error bars represent + -SEM ****$P < 0.0001$ as assessed by ANOVA. Experimental setup (B, C, D, E, F, G, H, I). Cells were incubated with DAMPs and analyzed for cholesterol load and uptake. **(C)** Representative filipin staining (green) in basal conditions (resting state) of the indicated spheroids (SP) and adherent (Adht) cells. The bar represents 0.5 mm. **(D)** Total cholesterol content (free and esterified) was measured using a Cholesterol/Cholesterol Ester-Glo assay kit. The measurements are relative to those in adherent cells. Mean values ± SEM of two independent experiments in replicate are shown. *$P < 0.05$, **$P < 0.01$, and ***$P < 0.001$ by two-tailed unpaired $t$ test. **(E)** Representative image showing filipin MFI values and peak shifts in unstained, control, and DAMP-treated spheroids and adherent cells. **(F)** Flow cytometry quantification of filipin III MFI values after DAMP treatment for 2 d of 3D and 2D-cultured indicated cells relative to solvent-treated controls. Data represent mean values ± SEM of two independent experiments. *$P < 0.05$, **$P < 0.01$, and ***$P < 0.001$ by two-tailed unpaired $t$ test. **(C, D, G)** Representative filipin staining (green) after 48 h of treatment with DAMPs (D) or solvent (C) of the indicated spheroids and adherent cells. DRAQ5 was used to label nuclear DNA (purple); the bar represents 0.5 mm. One representative image of at least two independent experiments is shown. **(H)** Total cholesterol load (free and esterified) was measured using a Cholesterol/Cholesterol Ester-Glo assay kit. Mean values ± SEM of two independent experiments in replicate are shown. *$P < 0.05$, **$P < 0.01$, and ***$P < 0.001$ by two-tailed unpaired $t$ test. **(I)** Total cholesterol quantification in P1 spheres treated for 48 h with DAMPs or solvent with or without simvastatin (5 $\mu$M). Mean values ± SEM of replicates in one experiment are shown. **(J)** Spheroids and adherent cells were incubated with fluorescently labeled cholesterol, treated with DAMPs or solvent for 30 h and labeled intracellular cholesterol load was determined

of PRRs with their respective ligands, including TLR2 by Pam3C, HMGB1, or HYLA; TLR3 by Poly(I:C); TLR4 by HS or LPS; and TLR9 by CpG and STING by 2'3 cGAMP. Whereas stimulation of individual TLRs did not markedly influence 3D-cultured EwS cell proliferation, stimulation of STING by cGAMP decreased proliferation of both 3D and 2D cultures of P1, P2, and C1 EwS cells (Fig 4B).

EwS are among the tumors with the highest levels of STING expression whose loss is associated with particularly poor prognosis (Fig S4D–F). Accordingly, STING was robustly expressed in both primary EwS cells and EwS cell lines and in human mesenchymal stromal cells, the putative cells of origin of EwS (Fig 4C). In contrast, cGAS was absent from the EwS cell lines in their resting state (Fig 4C) but could be induced by exposing the cells to various types of stress (data not shown), indicating that the gene had not been deleted or irreversibly repressed. Upon engagement by cGAMP at the ER, STING undergoes a conformational change as well as a variety of post-translational modifications associated with its dimerization and activation (21, 22, 23, 24, 25, 26). After activation, STING translocates from the ER to signaling compartments and associates with the Tank-binding kinase 1 (TBK1), which mediates the activation key transcription factors, including nuclear factor kB (NF-kB), that drive the expression of diverse cytokines, particularly type I interferon (22).

To gain insight into the effect of STING signaling on inflammatory mediator production in EwS cells, we assessed TBK1 and NF-kB activation and interferon-$\beta$ (type I) and -$\lambda$ (type III) expression by 3D and 2D cultured cells stimulated with cGAMP or DAMPs. Consistent with its effect in cells of the innate immune system, cGAMP-induced TBK1, and NF-kB activation, as assessed by TBK1 and p65 phosphorylation, respectively (Fig 4D left panel, Fig S4G). In contrast, DAMPs did not induce TBK phosphorylation, and generated only minor p65 phosphorylation (Figs 4D and S4G). Whereas cGAMP induced transient STING phosphorylation, DAMPs appeared to have no effect (Fig 4D right panel, Fig S4G right panel).

Because a major effect of STING is the induction of interferon type I and, more recently recognized, interferon type III (7, 27) expression, we assessed interferon-$\beta$ and $\lambda$ secretion by EwS cells stimulated with cGAMP. Consistent with its effect on cells of the immune system, cGAMPs induced a 1.5 to 8-fold and a 1.2 to 2-fold increase in secretion of IFN-$\lambda$ and IFN-$\beta$, respectively, by the different EwS used (Fig S4H–J). No increase in the secretion of type I and III interferons was observed in cGAMP-stimulated 293T cells, which do not express STING (Fig S4I) and P1 EwS cells produced no IFN $\beta$ because of a deletion of the corresponding gene (data not shown) (28). All of the EwS cells used expressed type I and III IFN receptors (Fig S4K). Stimulation of EwS cells by DAMPs caused a small increase in interferon $\lambda$ production that was partially blunted by DNAse, RNAs, and benzonase treatment, but not in interferon $\beta$ production, with the exception of a mild increase in C1 spheroids, as assessed by an IFN reporter assay, in both 3D and 2D-cultured cells (Fig S4L).

To determine the effect of STING signaling on EwS growth in vivo, A673 (C1) and TC252 (C2) cells were injected subcutaneously into NSG mice and tumors were allowed to reach a size of 50 mm³. cGAMP (10 $\mu$g in 50 $\mu$l) was then injected intratumorally (two injections over a 7-d span) and tumor size was assessed in comparison to tumors injected with solvent alone. Tumors were then removed, dissociated and the tumor cells subjected to RT-PCR and RNA-seq after removal of mouse cells (Fig 4E). Growth of both C1 tumors, whether derived from 3D or 2D cultures and tumors derived from 2D cultured C2 cells was inhibited by cGAMP injection (Fig 4F). To verify that the growth inhibitory effect of cGAMP was STING-dependent, C1 cells were depleted of STING using STING-specific shRNA. In response to cGAMP, growth of tumors derived from these cells was not delayed, whereas that of wt cells was strongly so (Fig 4G).

To determine whether the inhibitory effect of STING signaling on in vivo tumor growth might be due, at least in part, to IFN production, NSG mice bearing A673 (C1) tumors received two intratumoral injections of IFN-$\beta$ or IFN-$\lambda$ (50 ng or 25,000 IU each over a 7-d span) and tumor growth was compared with that of tumors injected with solvent only. Whereas both IFNs tended to delay tumor growth, the difference was not statistically significant (Fig 4H), consistent with the notion that they may participate in but do not alone explain the observed tumor inhibitory effect of STING signaling.

## Treatment of EwS cells with cGAMP causes changes in the expression of genes associated with lipid metabolism

Stimulation of STING by its natural ligand cGAMP caused delayed EwS growth in vivo as well as 3D and 2D cultured EwS cell growth in vitro. The growth inhibitory effect in vivo could only partially be recapitulated by intra-tumoral interferon $\beta$ and $\lambda$ injection (Fig 4H), leaving the mechanism responsible for the tumor cell-autonomous inhibitory effect of STING to be elucidated. To obtain clues as to the possible mechanism by which STING signaling might inhibit EwS cell growth, we conducted RNA seq on EwS cells stimulated with cGAMP in vitro as well as on cells derived from tumors that had been treated with cGAMP in vivo (Fig 5A). Using a twofold cutoff for gene expression change, we observed 519 and 256 up-regulated and 410 and 143 down-regulated genes in adherent C1 cells treated in vitro and in cells derived from tumors treated in vivo, respectively. The most highly represented GO terms for the up-regulated transcripts were Metabolic Process, Signal Transduction, and Biological Process (Fig S5A). Consistent with the growth inhibitory effect of STING signaling, overrepresentation of the terms negative regulation of cell proliferation, cell growth, and positive regulation of cell death appeared within the Biological Process term. Signal Transduction included enrichment of the terms response to type I interferon, innate immune response, cytokine mediated signaling pathway, and defense response. Interestingly, metabolic process

---

via a plate reader. Mean values ± SEM of three independent experiments in replicate are shown. *$P$ < 0.05, **$P$ < 0.01, and ***$P$ < 0.001 by two-tailed unpaired $t$ test. **(K)** Spheroids and adherent cells were incubated with cholesterol or mevalonic acid with or without simvastatin. Quantification of spheroid growth after 3 d of incubation in culture medium containing cholesterol (1 mM) or mevalonic acid (1 mM) with or without simvastatin (5 $\mu$M). Proliferation was assessed by MTT. Measurements are normalized to the solvent-treated control for each indicated cell line. Each experiment was performed at least three times. Mean values ± SEM are shown. *$P$ < 0.05, **$P$ < 0.01, and ***$P$ < 0.001 by two-tailed unpaired $t$ test. See also Fig S3.

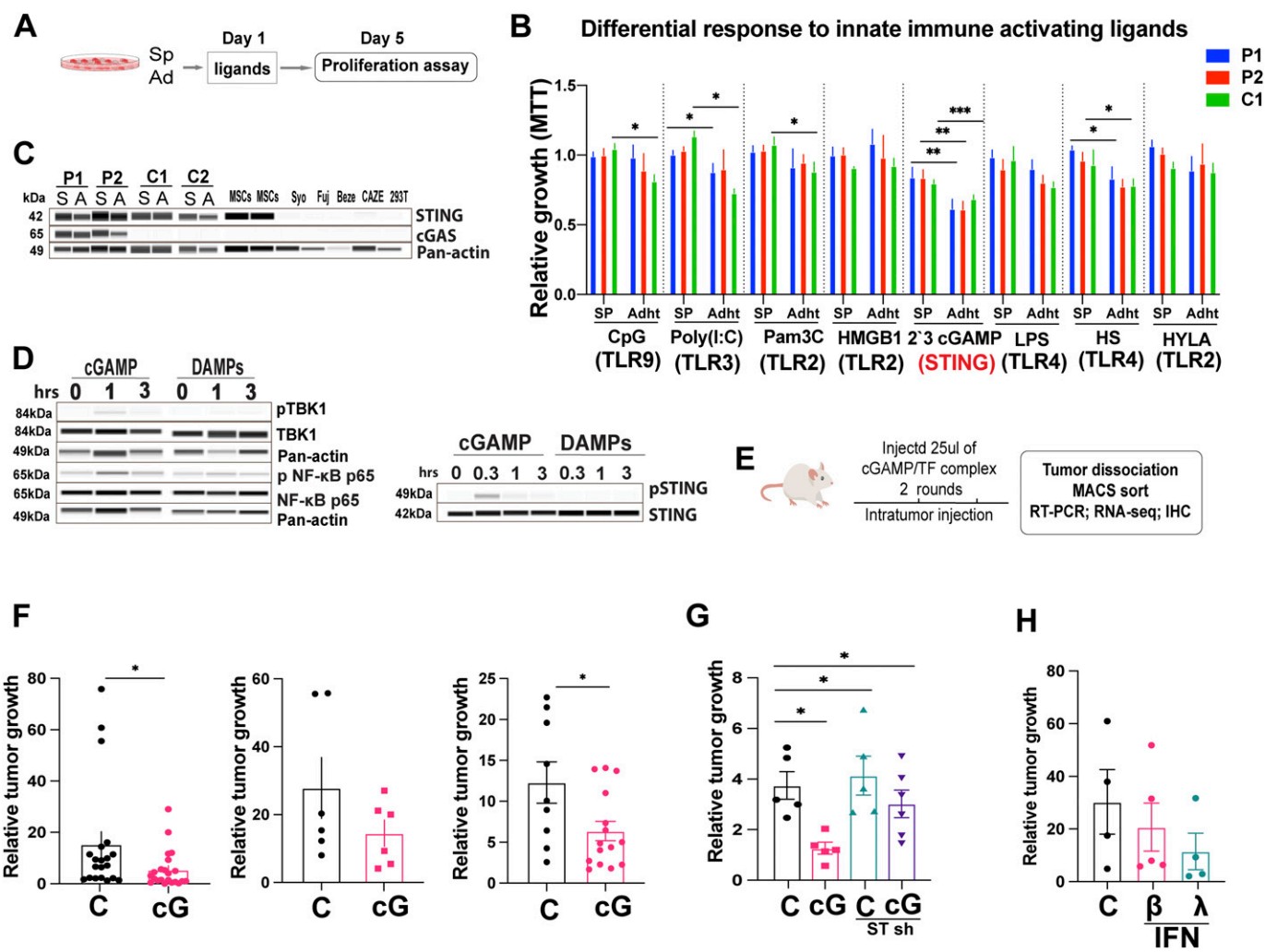

**Figure 4. Stimulation with cGAMP decreases EwS cell growth in vitro and in vivo.**
**(A)** Experimental setup. **(B)** Spheroids and adherent were incubated with the indicated TLR ligands and cGAMP for 5 d and proliferation was assessed by MTT and normalized to solvent-treated controls. The concentrations used were LPS (500 ng/ml), HMGB1 (1 μg/ml), CpG (2 μM), Pam3C (300 ng/ml), Poly(I:C) (10 μg/ml), 2'3 cGAMP (5 μg/ml), and HS (25 μg/ml) HYL (25 μg/ml). Each experiment performed at least three times in replicate. Mean values ± SEM are shown. *$P < 0.05$, **$P < 0.01$, and ***$P < 0.001$ by two-tailed unpaired $t$ test. **(C)** Expression of STING and cGAS, as assessed by Western blot analysis in Ewing sarcoma patient-derived spheroids and adherent cells (P1, P2) and in spheroids and adherent cells from EwS cell lines (C1, C2). Actin was used as the loading control. Each experiment was performed at least twice. **(D)** Phosphorylation of the indicated proteins was assessed by Western blot analysis after treatment of C1 adherent cells with 5 μg/ml 2'3'-cGAMP or damage-associated molecular patterns for the indicated duration. Images are representative of at least two independent experiments. **(E, F, G)** Experimental design for obtaining the data shown in (F, G) to assess the effect of cGAMP on EwS growth in vivo. NSG mice were injected subcutaneously with tumor cells and received intratumoral injection of solvent or cGAMP once the tumors had reached the size of 50 mm³ (day 0). Tumor volume at 15 d and was normalized to the volume at day 0 when cGAMP was injected. **(F)** Growth of tumors derived from adherent C1 cells (left panel), C1 spheroids (middle panel), and C2 adherent cells (right panel) injected with solvent as control (C, n = 20, 6, and 9, respectively) or 10 μg cGAMP (cG, n = 25, 6, and 15, respectively); injections were given on days 0 and 7. **(G)** Growth of tumors derived from adherent C1 cells bearing: non-silencing control shRNA (NSC) and injected with solvent (C, n = 5); NSC and injected with cGAMP (cG, n = 5); STING shRNA and injected with solvent (C, n = 5); and STING shRNA and injected with cGAMP, (cG, n = 6). **(F)** Injections were given as in (F) and tumor size was evaluated at 15 d. **(H)** Growth of tumors arising from adherent C1 cells injected with solvent (CTRL n = 4), IFNβ1 (50 ng (25000 IU), n = 5), or IFNλ1 (50 ng (25,000 IU), n = 4). Two injections were given on days 0 and 7 and tumor size was evaluated at 15 d. See also Fig S4.
Source data are available for this figure.

included enrichment of lipid metabolism pathways, including response to lipid, regulation of lipid metabolic process, lipid catabolic process, and lipid metabolic process. Enrichment of these terms was comparable in 3D and 2D-cultured cells (Fig S5A).

Several of the genes whose expression was modified by cGAMP in vitro and in vivo were associated with cholesterol homeostasis, but contrary to the effect of DAMPs on 3D-cultured cells, stimulation of STING appeared to down-regulate genes implicated in

cholesterol synthesis and up-regulate genes associated with cholesterol efflux (Fig 5B). GSEA plot analysis of C1 cells supported the notion that cGAMP up-regulates genes broadly related to cholesterol metabolism (Fig 5C). Remarkably, however, expression of genes implicated in the mevalonate pathway, which was up-regulated by DAMPs in 3D-cultured EwS cells, was down-regulated to 40–60% of control values by cGAMP (Fig 5D). Conversely, genes related to cholesterol transport and oxidation, including APOL1,

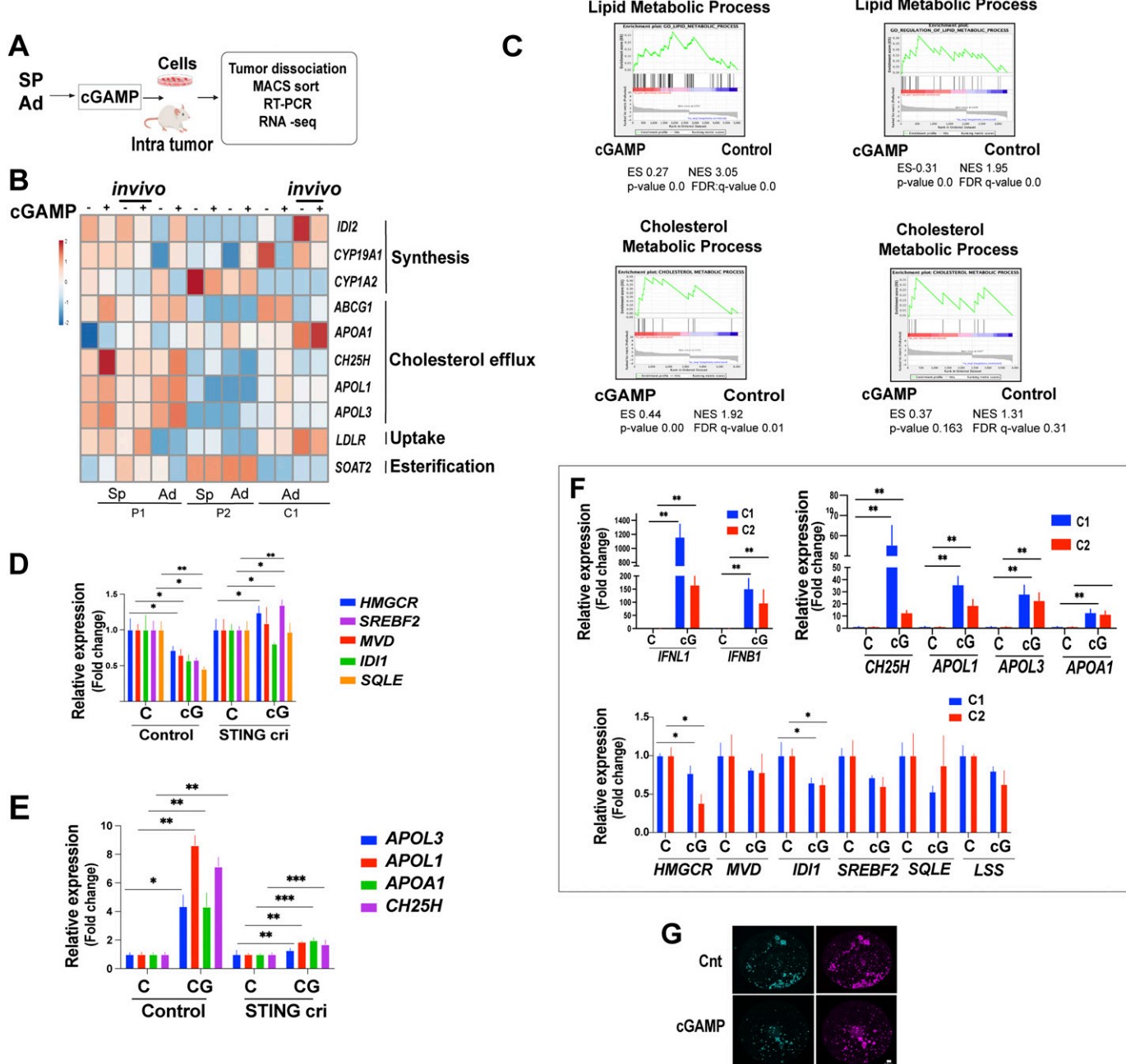

**Figure 5. Stimulation of EwS cells with cGAMP causes changes in the expression of genes associated with lipid metabolism.**
**(A)** Experimental setup (B, C, D, E, F). **(B)** Heat map showing the expression of selected genes implicated in cholesterol homeostasis from the RNA-seq data on spheroids (SP) and adherent cells (Ad) treated with cGAMP (4 h) in vitro and in vivo (marked as "in vivo"). **(C)** Gene Set Enrichment Analysis shows significant enrichment in lipid and cholesterol metabolic process genes upon cGAMP treatment of C1 adherent cells in vitro (left panels) and in vivo (right panels). **(D)** qPCR analysis of genes implicated in the cholesterol synthesis pathway in V2 control and CRISPR/Cas9 STING-depleted (cri) C1 spheroids treated with cGAMP (5 μg/ml) for 24 h, normalized to solvent-treated control. Each experiment was performed at least three times. Mean values ± SEM are shown. *P < 0.05, **P < 0.01, and ***P < 0.001 by two-tailed unpaired t test. **(E)** qPCR analysis of genes implicated in cholesterol metabolism/transport in V2 control and STING cri C1 spheroids treated with cGAMP (5 μg/ml) for 24 h normalized to solvent-treated controls. Each experiment was performed at least three times. Mean values ± SEM are shown. *P < 0.05, **P < 0.01, and ***P < 0.001 by two-tailed unpaired t test. **(F)** qPCR analysis of interferon and cholesterol metabolism and synthesis transcript levels in tumor cells derived from C1 and C2 tumors that had been injected with cGAMP (10 μg/ml) (24 h) in NGS mice. Data are compared with transfection complex control-treated tumors derived from the same cell lines. Data are reported as mean fold changes ± SEM after intrasample normalization to the levels of GAPDH for n = 3 animals per group *P < 0.05, **P < 0.01, and ***P < 0.001 (unpaired t test) compared with control-treated tumors of the same type. **(G)** Representative filipin staining (green) of C1 spheroids treated with cGAMP (5 μg/ml) for 2 d. DRAQ5 was used as to label nuclear DNA (purple) bar represents 0.5 mm. See also Fig S5.

APOL3, APOA1, and cholesterol-25-hydroxylase (CH25H) displayed a four to eightfold increase in expression (Fig 5E). Depletion of STING by CRISPR/CAS9 in C1 cells abolished the observed DAMP-mediated up-regulation of transcripts implicated in cholesterol transport as well as the repression of genes involved in cholesterol synthesis (Fig 5D and E).

Intratumoral injection of cGAMP in vivo had an even more marked effect. In addition to the robust induction of type I and type III interferons (Fig 5F, top left panel), expression of APOA1, APOL1, APOL3, and CH25H was increased from 10 to >50-fold, depending on the cell line (C1 or C2) from which the tumor originated (Fig 5F top right panel). Accordingly, expression of genes involved in cholesterol synthesis was decreased (Fig 5F bottom panel) as was the intracellular cholesterol load, as measured by filipin fluorescence (Fig 5G). Comparable observations were made on cells derived from P1 and P2 tumors (Fig S5B–D). Accordingly, the baseline level of STING expression correlated inversely with the expression of genes implicated in cholesterol synthesis and directly with the expression of transcripts associated with cholesterol transport (Fig S5E).

### STING and DAMPs exert opposing effects on cholesterol metabolism in EwS cells

Our observations thus far indicate that both STING activation and DAMPs exert a marked effect on the expression of genes implicated in lipid metabolism in EwS cells, and more particularly those associated with cholesterol synthesis, transport and oxidation. Remarkably, EwS appears to display the highest expression of genes implicated in cholesterol synthesis among a wide range of tumors, according to DepMap data (Fig S6A), hinting at the possibility that modulation of the cholesterol metabolism in EwS cells may have therapeutic relevance. To begin to address such a possibility, we compared the functional effects on EwS cell cholesterol uptake, efflux and load of stimulation or overexpression of STING to those of stimulation with DAMPs in the presence and absence of STING (Fig 6A).

To measure cholesterol efflux, control CRISPR/Cas9 (V2) and STING-specific CRISPR/Cas9-bearing C1 spheroids were incubated with fluorescently labeled cholesterol for 12 h, treated with cGAMP (5 μg/ml) for 24 h and their cholesterol efflux determined using a plate reader. Stimulation by cGAMP of C1 spheroids augmented their cholesterol efflux but failed to do so in spheroids depleted of STING (Fig 6B left panel). The effect was even more marked in adherent EwS P1 and C1 cells (Fig 6B middle panel) and was increased by STING overexpression in C1 cells bearing doxycycline-inducible STING (Fig 6B right panel). Total cellular cholesterol was increased at least threefold in DAMP-treated C1 spheroids depleted of STING (Fig 6C) but not in spheres overexpressing STING (Fig 6D).

The DAMP-mediated increase in total cellular cholesterol in cells depleted of STING was reflected by Filipin fluorescence (Fig 6E), whereas DAMPs failed to increase Filipin fluorescence intensity in cells overexpressing STING (Fig 6F). Cholesterol uptake was increased 1.2 to 1.5-fold in spheroids but not in adherent cells treated with DAMPs (Fig S6B).

Doxycycline-inducible expression of STING further illustrated its inhibitory effect on the expression of genes implicated in cholesterol synthesis and its stimulatory effect on the expression of genes associated with cholesterol oxidation and transport (Fig S6C and D). Finally, we observed that the genes implicated in cholesterol synthesis, including HMGCR, MVD, IDI1, SREBF2, and SQLE, which are induced by DAMPs, are down-regulated by β and λ interferons (Fig S6E). Conversely, both interferons augment the expression of APOL1, APOL3, APOA1, and CH25H (Fig S6F).

### STING tempers tumor growth and suppresses the tumor growth-enhancing effect of DAMPs

To address the potential cell-autonomous tumor suppressive effect of STING, we modulated STING expression in EwS cells and assessed the effect on xenotransplant growth in vivo. First, we depleted P1 cells of STING by shRNA (Fig S7A) and assessed their clonogenicity and tumor initiating ability in vivo along with the gene expression profile of the resulting tumors (Fig 7A). EwS cell clonogenicity was increased by STING depletion and decreased by STING overexpression (Fig S7B–D). Moreover, tumor growth from STING-depleted subcutaneous P1 EwS xenotransplants was markedly increased (P < 0.01) compared to that arising from control EwS cells and was accompanied by ovarian and liver metastases, which were not observed upon injection of cells expressing unrelated shRNA (Figs 7B and S7E). Gene expression profile analysis revealed that transcripts associated with lipid and cholesterol metabolism were strongly affected in EwS cells depleted of STING, with down-regulation of genes implicated in cholesterol transport and efflux (Fig 7C), consistent with our observations on the effect of cGAMP stimulation (Fig 5C–F).

We next compared the effect of DAMPs on tumor development by C1 EwS cells depleted of or overexpressing STING (Fig 7D). As observed above, DAMPs accelerated EwS tumor formation but the effect was significantly more pronounced when the tumor cells were depleted of IL29 (IFN-λ1) or STING (Fig 7E). Similarly, anti–IFN λ blocking antibody enhanced the growth promoting effect on EwS spheroids (Fig S7F). The effect was comparable whether STING depletion was achieved by shRNA or CRISPR/Cas9 (Figs 7F and S7A). GSEA plots revealed increased representation of genes implicated in lipid metabolism in EwS cells stimulated with DAMPs and a markedly accentuated representation of these genes and genes implicated in lipid transport and cholesterol metabolism in STING-depleted cells stimulated with DAMPs (Fig 7G). As expected, STING depletion abrogated cGAMP-mediated induction of its target genes (the top 50 target genes are shown in the heat map in Fig S7G) and augmented the representation of genes implicated in cholesterol biosynthesis (Fig S7H). Tumors resulting from EwS cells depleted of STING displayed decreased expression of genes implicated in cholesterol transport, as well as type I and III interferons (Fig S7I left panel) and a mild increase in genes implicated in cholesterol synthesis (Fig S7I right panel). Overexpression of STING neutralized the effect of DAMPs on tumor growth and mice injected with STING-overexpressing EwS cells survived longer than animals injected with control cells (Fig 7H). In all of the in vivo experiments, the depletion and overexpression of STING verified in cultured cells were maintained in the corresponding tumors (Fig S7J).

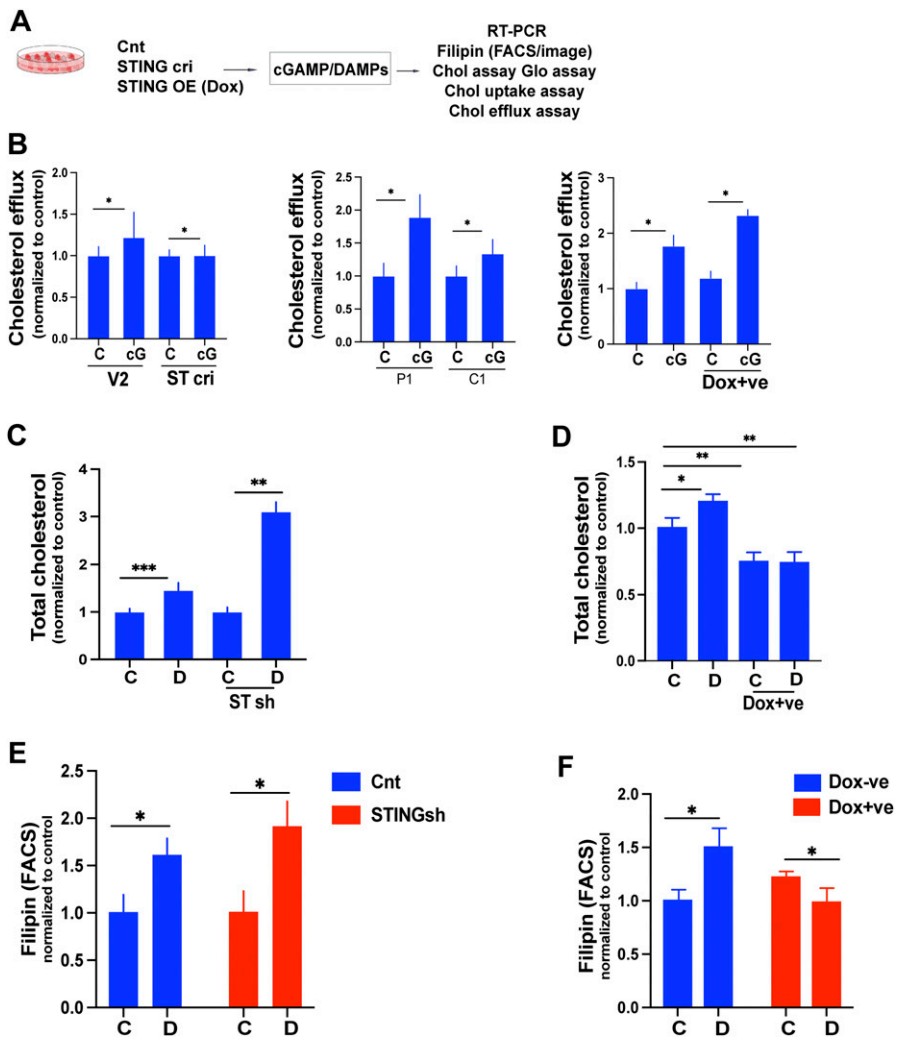

**Figure 6. STING and DAMPs exert opposing effects on cholesterol metabolism in EwS cells.**
**(A)** Experimental setup (B, C, D, E, F) Spheroids and or adherent were treated with cGAMP and analyzed for cholesterol efflux, total cholesterol load, and cholesterol uptake. **(B)** Cholesterol efflux was measured in V2 control and STING CRISPR (cri) C1 spheroids (left); P1 and C1 adherent cells (middle); and control (Dox −ve) and STING overexpressing (Dox +ve) C1 spheroids (right). **(C)** Cells were incubated with fluorescently labeled cholesterol for 12 h and subsequently treated with solvent (C) or cGAMP (cG, 5 µg/ml) for 24 h; cholesterol efflux was determined by assessment of the fluorescent label in conditioned culture supernatants via a plate reader. **(C, D)** Total cholesterol load was measured in control sh and STING sh C1 spheroids incubated with solvent (C) or DAMPs (D) for 48 h, using the Cholesterol/Cholesterol Ester-Glo assay kit. **(C, D)** Total cellular cholesterol was measured in V2 control (Dox −ve) and STING overexpressing (Dox +ve) C2 spheroids incubated with solvent (C) or DAMPs (D) for 48 h using the Cholesterol/Cholesterol Ester-Glo assay kit. Data were normalized to untreated controls. **(C, D, E, F)** Flow cytometry quantification of Filipin MFI values in control sh and STING sh cells treated with solvent (C) or DAMPs (D), (E); and V2 control (Dox −ve) and STING overexpressing (Dox +ve) C1 spheroids treated with solvent (C) or DAMPs (D), (F) for 2 d. Data represent mean values ± SEM of two independent experiments in replicate. *$P < 0.05$, **$P < 0.01$, and ***$P < 0.001$ by two-tailed unpaired $t$ test. See also Fig S6.

## STING and statins exert a synergistic inhibitory effect on EwS cell growth

Interrogation of the human metabolome database (HMDB), using the ENRICHER platform, with the genes whose induction was shared in DAMP treated and STING-depleted cells revealed Simvastatin to be the top drug/metabolite related to the gene product function (Fig S8A). These genes thus provide a signature predicted to be potentially targetable by statins. We therefore addressed the possible synergy between STING agonists and statins toward inhibiting EwS growth (Fig 8A). We observed that simvastatin together with cGAMP or STING overexpression markedly decreased EwS cell growth and viability (Fig 8B). Caspase 3 activity in EwS cells was mildly increased by cGAMP but more so in cells overexpressing STING (Fig 8C). Simvastatin alone did not increase caspase-3 activity in resting EwS cells or in cells stimulated with DAMPs but augmented the effect of cGAMP, which was further amplified in cells overexpressing STING (Fig 8C). Accordingly cGAMP alone decreased cell viability by roughly 20% but the effect, was amplified by simvastatin and the combination of STING

overexpression and administration of simvastatin (Fig S8B and C). As expected, the caspase-3 activity augmented by cGAMP and CH25H with and without simvastatin was decreased upon STING depletion (Fig S8D). The 15- to 30-fold induction of IFN λ and β expression by STING overexpression and cGAMP was amplified to more than 60-fold by statins, which alone did not induce IFN expression (Fig 8D), but tempered by an increased cholesterol load (Fig S8E). Consistent with these observations, total cellular cholesterol was decreased by statins and, to a lesser extent, by STING overexpression, with a marked synergistic effect between the two (Fig 8E).

To address the possible synergy between STING signaling and statins in vivo, we injected mice subcutaneously with C1 cells engineered to display doxycycline-inducible overexpression of STING. The mice were then divided into an untreated (native expression of STING) and a doxycycline-treated (STING overexpression) group and both groups received three injections of simvastatin IP (25 mg/kg), cGAMP IT (10 µg), a combination of both or solvent control on days 0, 3, and 7. Day 0 was defined as the time when the tumors had reached 150–200 mm³. Tumor size was

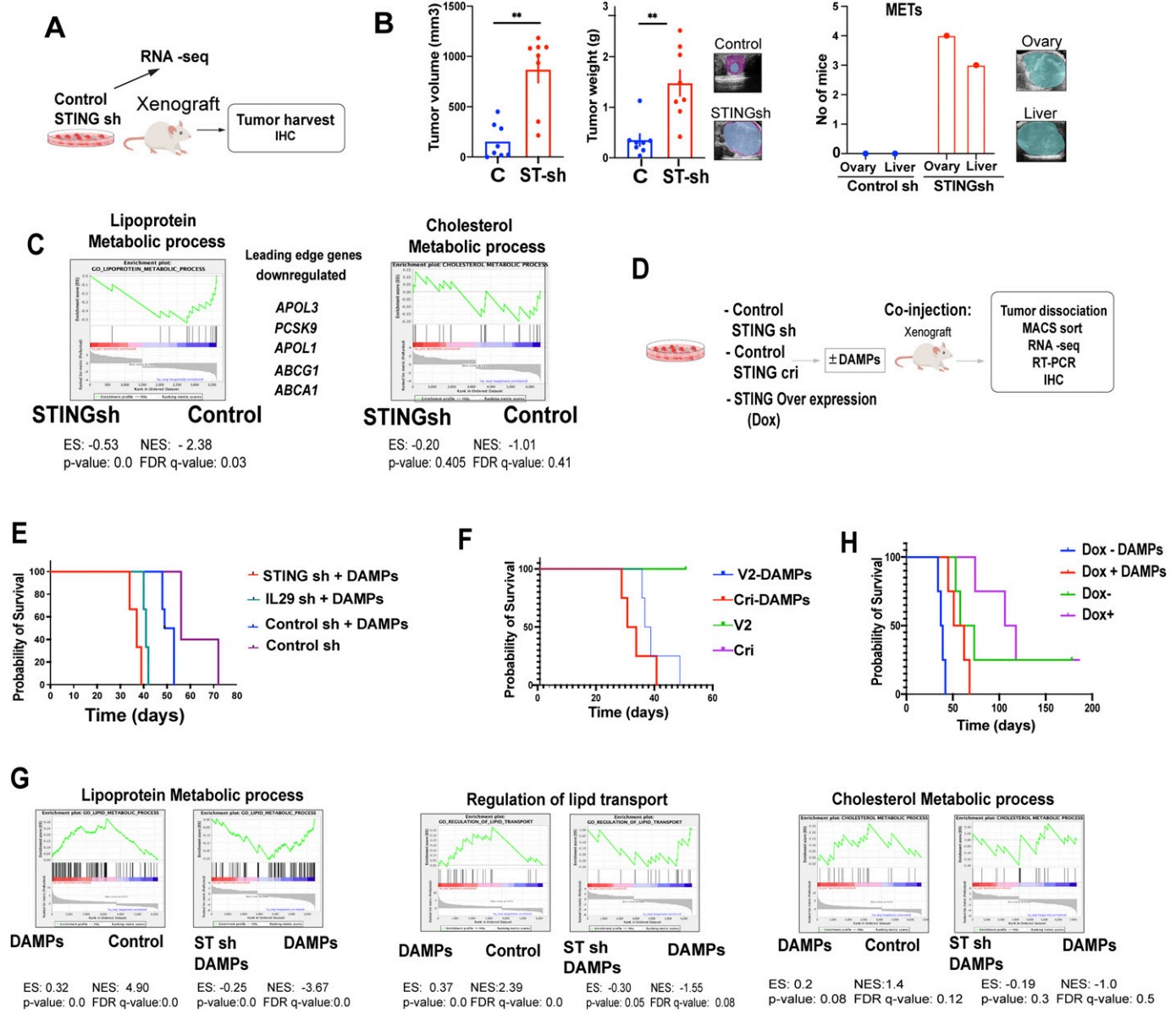

**Figure 7. STING tempers tumor growth and suppresses the tumor growth-enhancing effect of damage-associated molecular patterns (DAMPs).**
**(A)** Experimental setup for (B, C). **(B)** In vivo tumorigenicity of STING shRNA-depleted and control P1 spheroids in NSG mice (N = 8 mice/group). Left and middle graphs show tumor volume and weight, respectively. The graph on the right shows the number of mice with ovarian and liver tumor metastases, which were observed only upon STING deletion. Mean values ± SEM are shown. *P < 0.05, **P < 0.01, and ***P < 0.001 by two-tailed unpaired t test. **(C)** Gene Set Enrichment Analysis plots showing enrichment of the terms lipoprotein metabolic process (along with a list of leading-edge genes) and cholesterol metabolic process among twofold down-regulated genes in STING sh compared with control sh C1 adherent cells. **(D)** Experimental setup for (E, F, G, H). Control C1 spheroids and C1 spheroids depleted of STING by shRNA or CRISPR/Cas9 as well as spheroids overexpressing STING were injected into NSG mice and survival of the animals was measured (animals were euthanized when tumors reached 1 cm in diameter). Tumors were dissociated and the cells sorted and subjected to RNA seq, RT-PCR, and IHC. **(E)** Survival of mice injected with STING sh C1 spheroids + DAMPs, IL29 sh + DAMPs, control sh + DAMPs, and control sh C1 spheroids in the absence of DAMPs (n = 6). **(F)** Survival of mice injected with C1 STING CRISPR (cri) spheroids + DAMPs, C1 V2 control spheroids + DAMPs, C1 STING cri spheroids alone and V2 control C1 spheroids alone (n = 6). **(G)** Gene Set Enrichment Analysis plots depicting the enrichment of lipid and cholesterol-related Gene Ontology terms in twofold up-regulated genes in C1 spheroids treated with DAMPs compared with control cells, and twofold down-regulated genes in C1 STING sh spheroids treated with DAMPs for 4 h compared with control cells. **(H)** Survival of mice injected with STING overexpressing (Dox +ve) C1 spheroids + DAMPs; Control (Dox −ve) cells + DAMPs; STING overexpressing (Dox +ve) cells + solvent; and Control (Dox −ve) C1 spheroids + solvent (n = 6). Mean ± SEM are shown ****P < 0.0001 as assessed by log-rank (Mantel–Cox). See also Fig S7.

measured at day 12 following the initiation of treatment and normalized to the size at day 0. Simvastatin and cGAMP alone decreased tumor growth from cells with native STING expression and the combination of the two decreased growth even further

(Fig 8F). STING overexpression alone markedly decreased tumor growth and the decrease was bolstered by cGAMP administration. However, simvastatin had only a marginally incremental effect on tumor growth inhibition (Fig 8F).

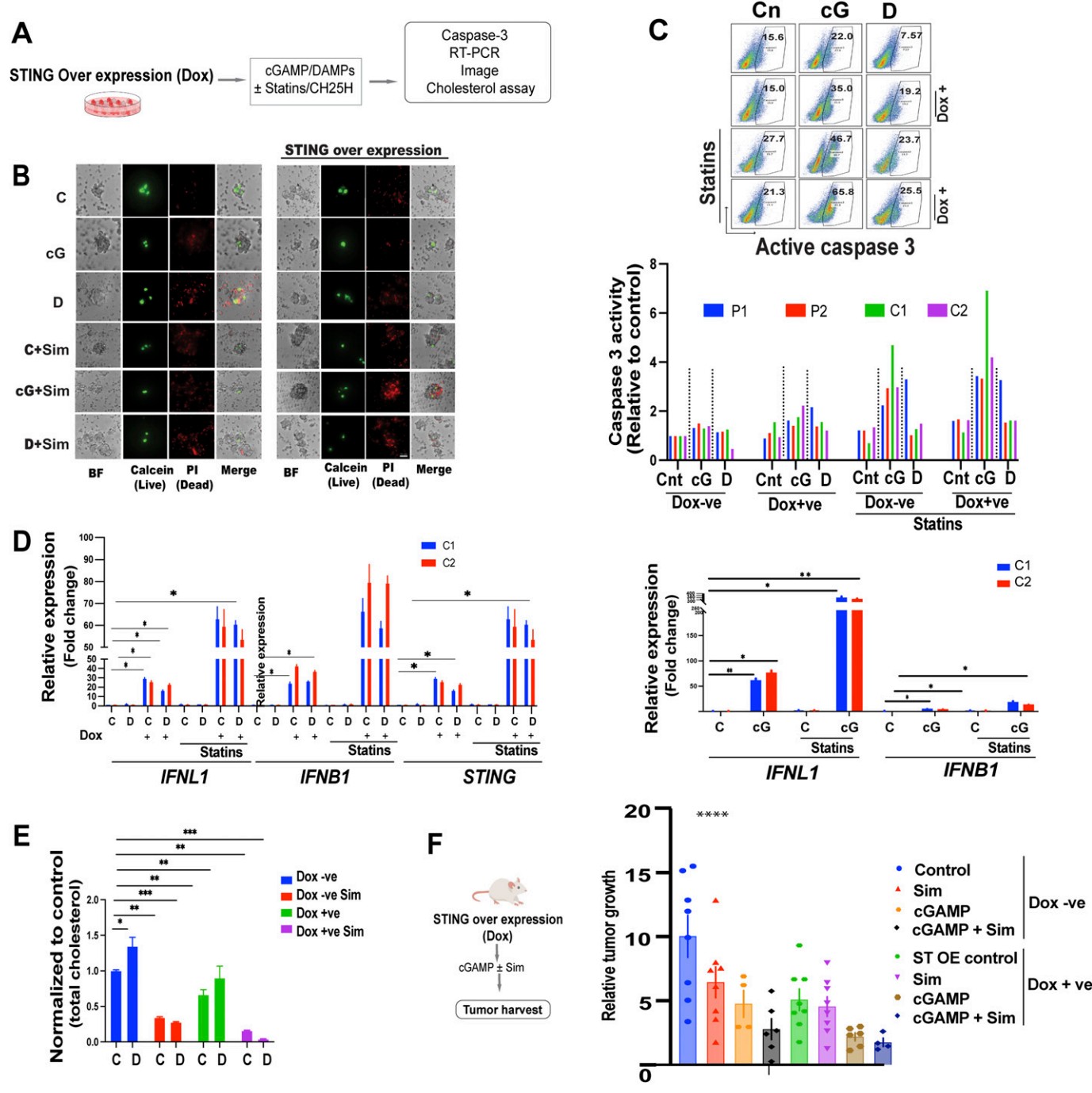

**Figure 8. STING and statins exert a synergistic inhibitory effect on EwS cell growth.**
**(A)** Experimental setup **(B, C, D, E)**. Control and STING overexpressing (Dox+) cells were treated with cGAMP, damage-associated molecular patterns (DAMPs), and statins and subjected to flow cytometry for analysis of active caspase 3, RT-PCR, and cholesterol load. **(B)** Fluorescence and light microscopy assessment of sphere formation by cGAMP (5 μg/ml); DAMPs ± statin- (5 μM) treated C1 spheroids. Staining with calcein (live), propidium iodide (dead) of Dox −ve control and STING overexpressing (Dox +ve) C1 spheroids assessed using a spinning disk confocal microscope 3 d after treatment. Scale bar 50 μm. **(C)** Flow cytometry analysis of cells for active caspase three in the indicated (Dox −ve) and STING overexpressing (Dox +ve) spheroids treated with cGAMP (5 μg/ml), DAMPs, and solvent ± statins (5 μM) for 3 d. Representative density dot plots (top) and quantification of active caspase 3 (bottom). Data were normalized to (Dox −ve) solvent-treated controls. **(D)** RT-PCR analysis of IFN-λ, IFN-β, and STING expression in (Dox −ve) and STING overexpressing (Dox +ve) C1, C2 spheroids 24 h after treatment with DAMPs or solvent ± statins (5 μM, right); RT-PCR analysis of IFN-λ and IFN-β expression in C1, C2 spheroids 24 h after treatment with cGAMP (5 μg/ml) and solvent ± statins (5 μM, left). Data were normalized to solvent-treated controls. **(E)** Total cellular cholesterol as measured using the cholesterol/Cholesterol Ester-Glo assay kit in (Dox −ve), and STING overexpressing (Dox +ve) C1 spheroids treated or not with DAMPs ± statins (5 μM) for 3 d. Data were normalized to (Dox −ve) solvent-treated controls. Mean values ± SEM of two independent experiments in replicate are shown. *P < 0.05, **P < 0.01, and ***P < 0.001 by two-tailed unpaired t test. **(F)** Experimental design to assess the possible synergy between simvastatin and cGAMP on EwS growth in vivo. NSG mice were injected subcutaneously with C1 spheroids containing vectors for Doxycycline-inducible STING overexpression. Once the tumors reached 150–200 mm³ (day 0), the mice received: (i) an intraperitoneal (IP) injection of 25 mg/kg simvastatin (InSolution TM Simvastatin, sodium salt, Merck - 567022); (ii) an

# Discussion

We have shown that DAMPs released by damaged cells undergoing necrosis can stimulate the growth of EwS spheroids as well as that of the corresponding tumors in vivo. In marked contrast, STING stimulation by its natural ligand cGAMP, had an almost diametrically opposed effect to that of DAMPs on EwS cell growth in vitro and on tumor growth in vivo and countered the growth stimulatory effect of DAMPs on EwS spheroids.

Upon interrogation of the possible mechanisms that underlie DAMP-mediated enhancement of EwS growth, we discovered that, similar to their effect on immune cells, DAMPs stimulate EwS cells to express a host of cytokines and chemokines implicated in inflammation. However, the repertoire of the induced mediators of inflammation was similar in 3D- and 2D-grown cells, whereas only the former displayed increased growth. The induced cytokines and chemokines therefore could not provide an explanation for DAMP-mediated enhancement of spheroid growth. In contrast, we observed a major difference in the transcription profiles, as related to genes implicated in lipid metabolism, between 3D and 2D cultured cells incubated with DAMPs. Culture conditions markedly affected EwS cell cholesterol metabolism and sensitivity to DAMPs. Thus, cells grown in 3D, which retain many of the in vivo properties that are lost in monolayer culture (29, 30), displayed higher cholesterol synthesis and uptake than cells grown in 2D. Similar observations were recently reported in breast cancer mammospheres (31) and in neuroblastoma neurospheres (32) suggesting that increased cholesterol uptake and/or synthesis, whose importance for tumor survival and progression is well established (19), may, together with a high SRC, be a general property of tumor spheroids and organoids that reflects tumor adaptability to stress in vivo. Tumor spheroids are enriched in tumor initiating/cancer stem cells (TICs/CSCs), which may, at least in part, be responsible for the observed increase in cholesterol synthesis (31).

Stimulation by DAMPs of EwS cells grown in 3D resulted in increased expression of genes associated with cholesterol synthesis and in increased cellular cholesterol load, whereas none of these changes occurred in cells grown as monolayers. Although the precise mechanism underlying the observed induction of cholesterol synthesis genes remains to be elucidated, it is likely that DAMPs contribute to the extrinsic, TME-derived, cues that stimulate tumor cell cholesterol metabolism. Such cues include hypoxia, a low pH due to increased glycolysis by cells composing the TME, inflammation and other types of stress (reviewed in reference 19). Incubation of EwS spheroids with increasing concentrations of cholesterol and, to a lesser extent, its precursor mevalonate in the culture medium stimulated proliferation, supporting the notion that the increased cholesterol load of 3D-grown cells in response to DAMPs provides a significant fraction of the fuel necessary to promote growth in vitro and in vivo. Taken together, these observations suggest that DAMPs resulting from spontaneous or drug-induced necrosis may increase the expression of numerous genes implicated in cholesterol uptake and synthesis, resulting in an increased cellular cholesterol load that stimulates EwS growth.

In contrast to DAMPs, stimulation of STING by its natural ligand cGAMP inhibited EwS cell growth in vitro, irrespective of the culture conditions and delayed the corresponding tumor growth in vivo. STING plays an important role in the innate immune response by controlling the expression type I and type III interferons, in addition to other mediators of inflammation (7). Interferons have long been known to inhibit tumor growth (33) by affecting the tumor cells directly and indirectly through stimulation of innate and adaptive immune responses in the TME (34, 35, 36). In Ewing sarcoma, interferons have been suggested to decrease tumor cell viability by engaging signaling pathways that induce apoptosis (37, 38). Accordingly, STING can inhibit tumor growth in a variety of settings (39, 40, 41), its anti-tumor effect being largely attributed to its induction of interferons.

Recently, a direct relationship between STING signaling and cell metabolism has been proposed (42). STING has been shown to associate with acetyl-CoA carboxylase (ACC) and fatty acid synthase in drosophila, and its deletion was observed to induce major metabolic dysregulation (42). Our observations indicate that STING activation by its natural ligand cGAMP reduces the cellular cholesterol load by up-regulating genes that encode proteins associated with HDL formation and cholesterol export from the cell. STING activation also induced genes encoding acyl-coenzyme A: cholesterol acyltransferases (ACATs, also known as SOATs) that generate cholesterol esters, which are stored in lipid droplets or secreted within lipoproteins (43). In contrast, cholesterol synthesis pathway genes up-regulated by DAMPs were repressed in EwS cells stimulated with cGAMP or overexpressing STING. Accordingly, the increase in proliferation of EwS spheroids induced by DAMPs was partially neutralized by cGAMP administration and STING overexpression. Consistent with these observations, STING depletion resulted in decreased expression of genes associated with cholesterol transport, increased expression of genes implicated in the cholesterol synthesis pathway, and an increased cellular cholesterol load. These effects were associated with stimulation of EwS cell proliferation and in vivo tumor growth as well as an increased growth promoting effect of DAMPs. The effect of STING signaling on the cholesterol metabolism of EwS cells could be recapitulated, at least in part, by type I and III interferons, indicating that in addition to a possible direct implication in the regulation of lipid metabolism, STING can modulate EwS cell cholesterol homeostasis via interferons. STING activity itself may be modulated by intracellular cholesterol levels, as incubation of cells in medium supplemented in cholesterol dampened the effect of cGAMP administration and STING overexpression on interferon induction. Conversely, limiting cholesterol synthesis alone induced type I interferon expression.

intratumoral (IT) injection of 10 µg cGAMP complexed with in vivo-jet transfection reagent; (iii) a combination of simvastatin IP and cGAMP IT (the dosage and timing of the injections were the same as for monotherapy); or (iv) solvent control (CTRL). Three injections were administered within a 7-d span (days 0, 3, and 7). The tumor volume on day 12 was normalized to the volume on day 0 when simvastatin, cGAMP, or combined treatment was begun. For Dox −ve animals: CTRL, n = 8; Sim, n = 8; cGAMP, n = 4; Sim+cGAMP, n = 6. For Dox +ve animals (tumors overexpressing STING): CTRL, n = 8; Sim, n = 8; cGAMP, n = 6; Sim+cGAMP, n = 4. Error bars represent + -SEM ****$P < 0.0001$ as assessed by ANOVA.

The sum of these observations suggests tight functional relationship between the innate immune response driven by STING and cholesterol homeostasis, which may have important implications in tumor growth. Because STING signaling reduced EwS cell proliferation and viability at least in part by decreasing their cholesterol load, STING agonists in combination with cholesterol depleting agents or inhibitors of cholesterol synthesis may have a robust anti-tumor effect in EwS.

Statins competitively inhibit 3-hydroxy-3-methylglutaryl coenzyme (HMG-CoA) reductase, the major rate-limiting enzyme that controls the conversion of HMG-CoA to mevalonic acid, thereby preventing mevalonate synthesis. They also exert potent anti-inflammatory activity, which may synergize with their inhibition of cholesterol synthesis toward limiting tumor growth. Statin use has been observed to be associated with decreased cancer-related mortality (44) and administration of statins as an anticancer strategy is being assessed in clinical trials (31). In breast cancer, increased cholesterol has been identified as a predictor of poor prognosis (45, 46) and inhibition of cholesterol synthesis has been found to provide beneficial effects in preclinical trials (47). In our present work, addition of simvastatin to culture medium containing increased cholesterol or mevalonate concentrations reduced their stimulatory effect on tumor cell growth. Simvastatin also inhibited the growth of EwS spheroids in the resting state and blunted the growth promoting effect of DAMPs. Statins enhanced STING expression and activity and the combination of STING overexpression and simvastatin administration had a strong synergistic effect toward inhibiting EwS cells growth in vitro. Similar effects were observed in vivo, as cGAMP and simvastatin together markedly reduced EwS growth. Interestingly, simvastatin did not increase the effect of cGAMP administration to STING overexpressing cells. A possible explanation, which remains to be verified, may be that modulation of cholesterol metabolism by stimulation of inordinately highly expressed STING reaches a threshold beyond which statins are no longer effective.

Currently, treatment of Ewing sarcoma is limited to conventional chemotherapy, which has reached a plateau in efficacy (11, 12), warranting the search for alternative therapeutic strategies. In light of the fact that EwS features among tumors with the highest expression of genes implicated in the cholesterol synthesis pathway and that it often presents extensive areas of necrosis even before chemotherapy, our observations suggest that STING agonists in conjunction with statins may offer an unsuspected approach to control EwS growth.

# Materials and Methods

## Primary tumor cells, cell lines, and 2D and 3D cultures

Establishment of the primary 3D EwS cultures (P1 and P2) and their derivation from EwS tumor samples was described previously (17, 18). Tumors were obtained from consenting patients with the approval of the ethics committee of the Canton de Vaud (Authorization No. 260/15). All human samples were anonymized before analysis and were exempted from informed consent by the law of the Canton de Vaud. Primary 3D (spheroid) tumor cell cultures were maintained in Iscove's modified Dulbecco's medium (Gibco) containing KO serum (20%; Gibco), recombinant human EGF (10 ng/ml; Invitrogen), recombinant human FGF (10 ng/ml; Invitrogen), and penicillin–streptomycin (1%; Gibco) in ultralow attachment flasks (Corning). Primary cultures ranging from 4 to 20 passages were used for experiments to minimize cellular drift. EwS cell lines used were A673 (ATCC) and TC252 (kindly provided by Dr. T. Triche, Children's Hospital Los Angeles, University of Southern California). Spheroids of A673 and TC252 cells were grown in ultralow attachment plates (Corning) under the same conditions as P1 and P2 cells. Adherent cells were derived from spheres and maintained in IMDM supplemented with 10% FCS (Gibco). Established cell lines were grown as monolayers in RPMI (Gibco) supplemented with 10% FCS. Cell lines were labeled C1(A673) and C2(TC252). Human embryonic kidney (HEK) 293T cells (ATCC) were cultured in DMEM (Gibco) supplemented with FBS (10%; PAN-Biotech), MEM nonessential amino acid (Gibco), and penicillin–streptomycin (1%; Gibco) supplemented with 10% FCS (Gibco). Cell cultures were maintained at 37°C and 5% $CO_2$ in humidified culture incubators.

## Induction of necrotic and apoptotic cell death

To obtain DAMPs from necrotic cell death, tumor cells were harvested, resuspended in water, at a concentration of $10^7$ cells/500 $\mu l$, and subjected to 3–5 freeze–thaw cycles. Lysates were centrifuged at 13,000$g$ for 10 min, the pellets containing cellular debris were discarded and DAMP-containing supernatants were collected and used immediately for further experiments. Necrosis was confirmed by trypan blue and annexin V–PI staining by flow cytometry. The presence of DNA in DAMP-containing supernatants was verified using the PicoGreen assay. To generate apoptotic bodies, tumor cells were treated with camptothecin (3 $\mu$M) (C-9911; Sigma-Aldrich) for 24 h. Apoptosis was scored using annexin V–PI staining.

### Nuclease treatment

Necrotic cell supernatants were treated with DNase 1 $\mu$/ml (79254; Sigma-Aldrich), RNaseA 1 $\mu$l/ml (EN0531; Thermo Fisher Scientific), or Benzonase 1 $\mu$l/ml (70746; Novagen) at 37°C for 30 min. Cells were then incubated with nuclease-treated supernatants (3 d for proliferation assays; 24 h for IFN assessment) and their proliferation (MTT) and IFN expression and activity (RT-PCR and Luminex, respectively) were analyzed.

## Oxygen consumption rates

For 2D culture respiration assays, cells were plated in growth media at a density of 50,000 cells per well in XF96 microplates (102601-100; Seahorse Bioscience) 24 h before the assay. Growth medium was changed to XF assay medium (103576-100; Seahorse Bioscience), and the plates were incubated at 37°C in a non-$CO_2$ incubator for 45 min before starting the assay. To measure respiration in spheroids, spheroids were grown for 2 d in 96-well Ultra-Low Attachment round-bottom plates starting with 50,000 cells per well (102959-100; Seahorse Bioscience), resulting in an average spheroid diameter of 200 ± 10 $\mu$m. The next day, spheroids were transferred to a Poly-d-lysine-treated XFe96 Spheroid Microplate (Cat. no. 102959-100; Seahorse Bioscience) containing 37°C XF

assay medium. Seahorse XF Cell Mito Stress Tests were carried out according to the manufacturer's protocol using sequential injections of Oligomycin (1.5 $\mu$M), FCCP (1 $\mu$M), and Rotenone plus Antimycin A (0.5 $\mu$M). For consistency, we used the same concentrations for both spheroids and adherent cells. To normalize the data, the cells were labeled with DAPI, images acquired, and cell counts quantified using Cytation 3 Cell Imaging Reader (BioTek). SRC was calculated as the maximal–basal oxygen consumption rates.

## Tumor growth in vivo

EwS spheroids and adherent cell lines containing no exogenous vectors as well as those expressing control shRNA, STING shRNA, V2 CRISPR control sequences, STING CRISPR sequences (cri), or a doxycycline-inducible STING expression plasmid were injected subcutaneously (200,000 cells) or beneath the kidney capsule (10,000 cells) of 6–8-wk old female NSG mice. For STING overexpression, mice were fed a doxycycline diet ((SAFE Nutrition Service: 150–1 g/kg)). All animal experiments were conducted according to the guidelines of the Veterinary Service of the Canton of Vaud under the authorization number VD2488.2.

Subcutaneous tumor growth (C1, C2) was monitored by measuring the tumor diameter with a caliper every 3 d until it reached 1 cm. Tumor volume was calculated using the formula: Volume (mm$^3$) = (l × w$^2$)/2, where w (width) is the smaller and l (length) is the larger of two perpendicular tumor axes. For kidney capsule injection of P1 and P2 cells, tumor growth was monitored by ultrasound imaging using a 40-MHz probe and the Vevo 2100 Ultrasound machine (VisualSonics). Tumor volume was calculated by V = 1/4 4/3 p(Dd Ds Dt)/8, where Dd corresponds to tumor height, and Ds and Dt to tumor lengths measured along long- and short-axes, respectively. When control tumors reached 1 cm$^3$, mice were euthanized.

Tumors were harvested at autopsy and processed for RNA extraction, H&E staining, immunohistochemistry (IHC) and, when possible, 200 mg of tumor tissue were enzymatically dissociated for MACS sorting by gentle MACS dissociator (130-093-235; Miltenyi Biotec). Red blood cells were removed by RBC lysis buffer (130-094-183; Miltenyi Biotec) and mouse cells were depleted using a mouse cell depletion kit (130-104-694; Miltenyi Biotec) and anti-mouse CD45 MicroBeads (Cat. no. 130-110-618; Miltenyi Biotec). Sorted tumor cells were subjected to gene expression analysis by RNA seq and qRT-PCR.

## Stimulation of EwS cells with DAMPs and apoptotic bodies in vitro and in vivo

For stimulation in vitro with DAMPs, cells were incubated with necrotic cell supernatants at a 1:20 dilution at 37°C. For stimulation in vivo, necrotic cancer cell supernatants were added to cell suspensions at final dilution of 1:10 and the mixture was incubated for at least 30 min on ice. Control tumor cells were treated identically with substitution of the necrotic cell supernatant by an equal volume of water. For stimulation in vitro with apoptotic bodies, cells and apoptotic bodies were co-cultured directly at a 1:10 volume ratio of apoptotic body preparations to living cell suspensions at 37°C. For stimulation in vivo, apoptotic body preps and living cell suspensions were mixed at a 1:5 ratio. The suspension mix was injected subcutaneously or beneath the kidney capsule of 6–8-wk-old female NSG mice. Subcutaneous injections included 1 × 10$^6$ adherent or 2 × 10$^5$ spheroid-derived cells/injection. Kidney capsule injections consisted of 10,000 spheroid-derived cells.

## In vivo administration of cGAMP and IFNs

For cGAMP and IFN treatment in vivo, a total of one million adherent cells or 200,000 spheroid-derived cells were injected subcutaneously into the flank of 6-wk-old female mice. Once tumors reached 50 mm$^3$, two intratumoral injections of solvent (control) or 10 $\mu$g cGAMP (Invivogen) complexed with in vivo-jet (in vivo-jetPEI) transfection reagent were given within a 7 d span; for IFN, two intratumoral injections of solvent (control) or 50 ng (25000 IU) IFN$\lambda$ (300-02L) or IFN $\beta$ (300-02BC; Pepro Tech) were given within a 7 d span. In all cases, the tumor volume was normalized to the volume at day 0.

## In vivo tumor treatment using simvastatin and cGAMP in the presence or absence of STING overexpression

One million cells containing the Doxycycline-inducible STING expression plasmid were injected subcutaneously into 6–8-wk-old female NSG mice. Treatment was started once tumors had reached 150–200 mm$^3$. Mice were divided into Dox –ve and Dox +ve groups. For STING overexpression (Dox +ve), mice were fed a doxycycline diet ((SAFE Nutrition Service: 150–1 g/kg)). Intraperitoneal injection of 25 mg/kg simvastatin (InSolution TM Simvastatin, sodium salt, 567022; Merck), intra-tumoral injection of 10 $\mu$g cGAMP complexed with in vivo-jet transfection reagent, and combination treatment (the dosage and timing were the same as for monotherapy) were given within 7 d (days 0, 3, 7). In all cases, the tumor volume was normalized to the volume at day 0.

## Clonogenicity and cell proliferation

Cell proliferation was assessed by MTT 5 d after plating cells of interest at a density of 5,000 cells/well in a 96-well plate (G5430; Promega, Cell-Titer Glo reagents [Promega-G7570]). Six technical replicates were done of each proliferation experiment and values are presented relative to those of viable vehicle-treated cells, which are set as one. For clonogenicity experiments, cells were sorted using a flow cytometer (BD FACSAria II) into 96-well plates at a density of 1, 5, 10, 25 live cells per well and maintained in culture for 3 wk before sphere formation was scored. All spheroid formation and proliferation experiments were performed at least twice.

## PRR ligands, lipid signaling modulators, and cytokines/chemokines

PRR ligands and their concentration of administration were as follows: 2'3 cGAMP 5 $\mu$g/ml (TLRL-NACGA23), CpG 2 $\mu$M (TLRL-2216), Pam3C 300 ng/ml (tlrl-pms), and poly I:C 10 $\mu$g/ml (tlrl-picw) were all from Novogen. HS 25 $\mu$g/ml (GAG-HS I; Iduron), HYL-25 $\mu$g/ml (HAO2; Iduron), LPS 500 ng/ml (L2880; Sigma-Aldrich), and HMGB-1 $\mu$g/ml

(Z02803-50; Genscript). Lipids/lipid signaling modulators and their concentration of administration were cholesterol, 1 mM (HY-N0322; MCE med chem express); mevalonic acid, 1 mM (HY-113071A; MCE med chem express); and CH25H, 10 $\mu$M (HY-113134; Med chem express). The chemokines and cytokines CXCL10 (300-12), CCL5 (AF-300-06), IL1$\beta$ (200-01B), and CCL2 (AF-300-04), all from Pepro Tech, were administered at 10, 50, and 100 ng/ml. Cells were treated with each reagent for 4 or 5 d and their growth was assessed by MTT or Cell-Titer Glo reagents.

For cholesterol synthesis inhibition, cholesterol depletion, and IFN signaling blockade, cells were treated for 24 h or 3 d, respectively, with simvastatin (5 $\mu$M) (S6196; Sigma-Aldrich), MBCD (5 mM) (C4555-MCE Med chem express), and human anti-IL-29 antibody (mabg-hil29-3; InvivoGen) was added at the final concentration of 100 ng/ml for 1 h before any further treatment.

### Type I and III IFN reporter and Luminex assays

Interferon activity (IFN-$\alpha$, IFN-$\beta$, and IFN-$\lambda$) was monitored using HEK-Blue IFN-$\alpha$/$\beta$ (hkb-ifnab) and HEK-Blue IFN-$\lambda$ (hkb-ifnl) reporter cells (Invivogen). Primary EwS cells and EwS cell lines were treated with 5 $\mu$g/ml of cGAMP, DAMPs (1:10 dilution of necrotic cell supernatants), 1 mM cholesterol, 1 mM mevalonic acid, 5 $\mu$M simvastatin, or solvent control for 24 h. Supernatants were then collected, incubated with the reporter cells, and IFN activity assessed according to the manufacturer's instructions. Luminex assays were performed using a bead-based immunoassay (human custom Procarta Plex [ppx-02-MXDJX4X; Invitrogen]) according to the manufacturer's instructions.

### Cholesterol load, uptake, and efflux

Cellular cholesterol was quantified using the J3190-Promega Cholesterol/Cholesterol Ester-Glo kit according to the manufacturer's instructions. Briefly, 5,000 sphere-derived or adherent cells were plated per well in triplicate in 96-well plates for 6 h. Cells were stimulated with cGAMP, (5 $\mu$g/ml) DAMPs (1:10 dilution), simvastatin (5 $\mu$M), or solvent control for 48 h. 50 $\mu$l of cholesterol lysis solution was added to each well and incubated for 30 min at 37°C. Cholesterol detection reagent and cholesterol detection reagent with esterase for both samples and standards were prepared according to the manufacturer's instructions in a final reaction volume of 100 $\mu$l per well.

Filipin staining was done using the cholesterol assay kit (ab133116; Abcam). For fluorescence microscopy, sphere-derived and adherent cells were seeded at a density of 1,000 per well of a 96-well plate and treated with cGAMP (5 $\mu$g/ml), DAMPs, or simvastatin (5 $\mu$M) for 48 h and the staining assessed using a Leica DMi8 microscope, equipped with an Andor Zyla Fluotar camera, at 10×/0.32 magnification. Images were analyzed by 1× Leica Application Suite X software. For flow cytometry, sphere-derived and adherent cells were seeded at a density of 100,000 per well of a 24-well plate and cGAMP (5 $\mu$g/ml), DAMPs, simvastatin 5 $\mu$M, or solvent control were added and maintained for 24 h. Flow cytometry was performed using a Fortessa 2 (BD Biosciences) and the data were analyzed using FlowJo.

The cholesterol uptake assay was conducted according to the manufacturer's instructions (ab236212; Abcam). Cells were seeded at a density of 5,000 per well of 96-well plates and incubated with 20 $\mu$g/ml NBD Cholesterol with or without cGAMP (5 $\mu$g/ml) for 30 h.

The cholesterol efflux assay performed according to the manufacturer's instructions (ab196985; Abcam). THP1 positive control cells treated with 100 nM phorbol 12-myristate 13-acetate (PMA) for 48 h to optimize the condition for cancer cells. Briefly, control, STING-depleted, and STING overexpressing cells were seeded in 96-well plates (10,000 cells per well), maintained at 37°C for 24 h, then washed in serum-free RPMI and incubated with labeling reagent overnight. Cells were then incubated in fresh serum-free RPMI with or without cGAMP (5 $\mu$M/ml) for 24 h. Culture medium and cell lysate cholesterol efflux was calculated based on the fluorescence intensity of the medium and that of cell lysates according to the manufacturer's instructions.

### Flow cytometry

#### *Apoptosis assay*
Spheroid-derived cells were seeded at a density of 80,000–100,000/well of a 24-well plate and cGAMP 5 $\mu$g/ml, CH25H, simvastatin (5 $\mu$M), or solvent control were added and incubated for 24–72 h. Cells were stained with annexin V/PI(FITC Annexin V Apoptosis Detection Kit II [556420; BD Biosciences]), anti-active caspase 3 antibody (550480; BD Biosciences) or calcein AM cell-permeable-dye (C1430; Thermo Fisher Scientific), and flow cytometry was performed using an LSRII Fortessa flow cytometer. Data were analyzed using FlowJo.

Cell surface IFN receptor (IFNAR1/IFNAR2/IFNLR1) expression was assessed using anti-IFNAR1 (FAB245P; R&D Systems), anti-IFNAR2 (130-099-560; MILTENYI BIOTEC GMBH), and anti-IFNLR1(337803; BioLegend) antibodies. Cells were incubated with the antibodies for 30 min at RT, washed, and analyzed using a Gallios Cytometer (Beckman Colter).

### Microscopy

Spheroid-derived and adherent cells were seeded at a density of 1,000/well of a 96-well plate with and without doxycycline. After 24 h, cGAMP, DAMPs, statins, or solvent control were added to cells and incubated for 48 h. Cells were then labeled with calcein-AM and PI to identify live and dead cells, respectively. Images were taken with a Nikon Eclipse microscope Ti2!x Nikon DS-QI2, at Plan Fluor 40× oil N.A 1.30 magnification and analyzed by 1× Nikon DS-Qi2 software.

### Immunohistochemistry

Formalin-fixed, paraffin-embedded tumor tissue was sliced into 8-$\mu$m sections and subjected to standard H&E staining. Spheroids and adherent cells were re-suspended in 1.5% low-melting agarose and included in paraffin blocks. The blocks were cut, sections mounted onto slides and stained using anti-STING and anti-cGAS antibodies. Images were taken with a Hamamatsu NanoZoomer sCMOS Orcs Flash LT plus microscope analyzed by NDP.view2plus software.

## RNA extraction and real-time PCR

Total RNA was extracted using the RNeasy Mini Kit (QIAGEN), following the standard manufacturer's protocol. Real-time PCR amplification was performed using Power SYBR Green PCR Master Mix (Applied Biosystems) in a QuantStudio 5 System instrument (Thermo Fisher Scientific). Primer sequences used for gene expression are listed in Table S1. Primers were selected according to PrimerBank. Relative quantitation of gene expression was conducted according to the $2^{-\Delta\Delta Ct}$ method.

## Lentiviral infection of shRNA, CRISPR, and overexpression constructs

For knockdown (KD) of STING, lentiviral shRNA plasmids (V3LHS-324286, V3LHs 324289, and V3LHS-324284) were obtained from Dharmacon. A non-silencing vector containing a scrambled sequence was used as a control (ref. RHS4346, Dharmacon). The STING CRISPR plasmid and V2 CRISPR control were a generous gift from Fabio Martinon. The Doxycycline-inducible STING overexpression plasmid was kindly provided by Andrea Ablasser (48). To induce STING overexpression, cells were treated with 0.5 μg/ml doxycycline (Sigma-Aldrich).

Fore lentiviral production, HEK 293T cells (ATCC) were transfected using FuGENE 6 Transfection Reagent (Promega). pMD2G (#12259; Addgene) and pCMVΔR8.74 (#12263; Addgene) vectors were used as envelope and packaging plasmids. Lentiviral harvest was performed using Lenti-X Concentrator (TAKARA). Cells were infected with plasmids indicated above for 16 h and selected with 1 μg/ml of puromycin (Invivogen) for 48 h before further analysis.

## Western blots

Cell pellets were resuspended in SDS lysis buffer (50 mM Tris, pH 6.8, 2% SDS, 5% glycerol, two mM DTT, 2.5 mM EDTA, 2.5 mM EGTA, 4 mM $NaVO_4$, and 20 mM NaF complemented with Complete Protease inhibitors and PhosStop [Roche]) and sonicated for 10 s with 2 s ON and 2 s OFF. After 5 min at 95°C and centrifugation for 5 min at $10,000g$, supernatant proteins were quantified using the tryptophan fluorescence method (Wisniewski and Gaugaz (2015) Anal Chem 87: 4110–4116). 1.2 μg of protein extracts were subjected to the capillary-based immunoassay Jess system (Protein Simple) using the 12–230 kD separation module according to the manufacturer's instructions. Anti-STING (13647S), anti-cGAS (15102), anti–NF-κB p65 (6956), anti–phospho-NF-κB p65 (3033), anti–phospho-STING (Ser366) (19781), anti-TBK1 (3504), anti–phospho-TBK1 (5483), and anti–pan-Actin (4968) antibodies were purchased from Cell Signaling Technology and diluted 40-fold in the AD2 blocking buffer before use.

## Statistical analysis

Statistical tests and graphics were generated by Prism version 7.03 (GraphPad Software Inc.) and excel. An unpaired, two-tailed $t$ test and log-rank (Mantel–Cox) test analysis were used to determine the significance of the difference between the control and each experimental condition. $P$-values < 0.05 were considered statistically significant and are denoted by asterisks: *$P$ < 0.05, **$P$ < 0.01, and ***$P$ < 0.001. Error bars represent the SEM.

## Bioinformatics analysis

### RNA-seq data

Total RNA was extracted using the RNeasy Mini Kit (QIAGEN), and the sequencing library was prepared using the Illumina TruSeq library preparation kit following the manufacturer's instructions. The libraries were sequenced on the Illumina HiSeq 4000 platform as single-end 100-bp reads (IGE3 genomic platform of the University of Geneva). Spheres and adherent cells were treated with cGAMP or water for 4 h, whereas for DAMPs treatment, cells were treated for 4 or 24 h as mentioned in the sample description. The reads were first QC-checked using FastQC and then mapped on human genome hg38 using STAR (v2.5.2b; REF1). The mapping of the sample "P1 spheroids control—in vivo (A1)" was not satisfactory (only 60% reads aligned uniquely versus around 90% for other samples) and it was re-sequenced with 50-bp reads. All samples were analyzed and normalized together using DESeq2 (v1.20.0; REF2). A minimum filtering was applied (total number of counts > [number of samples/2]). Normalized counts were obtained by dividing the actual counts with the respective size factors obtained from the *estimateSizeFactors* function of DESeq2. Differential analysis fold-change was obtained by dividing the normalized counts in respective samples. Because there were no replicates, it was not possible to calculate the $P$-values. For the heat map of DAMPs, only the DAMP samples were normalized together. Then only the protein-coding genes were considered, and the top 10 genes according to the fold change of P1 DAMPs versus P1 control were plotted on the heat map.

### GSEA

GSEAPreranked (GSEA v4.1 (49, 50)), as suggested in GSEA FAQ for RNA-seq data, was used to run the GSEA analyses. For GSEA analyses, only genes showing twofold up- or down-regulation were considered. The GO enrichments were obtained using the GSEA tool using "C5: ontology gene sets" of MSigDB collection.

### DepMap gene expression boxplots

Expression data (21Q3 Public) was downloaded from DepMap portal (51). Expression data were used to generate boxplots for different cancer types. Bone cancer data were subdivided into Lineage Subtypes "Ewing Sarcoma" and "Bone Cancer." Only Cancer types having at least two samples were used for plotting. Fibroblast and Engineered data were also removed.

### Survival analysis

According to the expression of STING, survival analysis was performed using the R2 Genomic portal; data chosen was "Mixed Ewing Sarcoma - Savola - 117 - MAS5.0 - u133p2." Expression levels were categorized into two groups: first quartile versus rest other by choosing the cut-off mode as "the first quartile." Kaplan–Meier curve was plotted using overall survival data.

### Inflammation and cholesterol gene sets

A list of 204 genes involved in cholesterol biology was derived using the following GO ontologies: "GO:0006695" = cholesterol

biosynthetic process "GO:0042632" = cholesterol homeostasis, "GO:0045540" = regulation of cholesterol biosynthetic process, and "GO:0008203" = cholesterol metabolic process. The inflammation gene set was obtained from the H collection of The MSigDB database of GSEA with the name: "HALLMARK_INFLAMMATORY_RESPONSE."

### Cholesterol, fatty acids, and lipid metabolism pathways

The genes for pathways were obtained from the REACTOME database (52). In particular, the following pathways for cholesterol biosynthesis and lipid metabolism were downloaded: "cholesterol biosynthesis," "fatty acid metabolism," "phospholipid metabolism," "regulation of cholesterol biosynthesis by SREBP (SREBF)," and "sphingolipid metabolism."

### Enrichment analysis

HMDB metabolites enrichment and KEGG 2021 enrichment were done using the Enrichr tool using twofold up-regulated genes (53).

### Ewing sarcoma expression data and low-high boxplots

Expression data were downloaded from GEO (Accession ID: GSE34620). This dataset contains 117 Ewing sarcoma samples profiled using Affymetrix hgu133Plus2 arrays. Cell files downloaded from GEO are RMA-normalized to get the final expression values. The expression values of each sample were scaled using the *scale()* function of R to get a z-score. The expression values of any particular gene across all samples were then categorized into three categories according to low (STING expression < first Quartile), middle (STING expression between first and third Quartile), or high (STING expression > third Quartile). The *P*-value between the High and Low categories was calculated using a two-tailed *t* test. An average z-score was used for all the genes involved in pathways.

## Data Availability

The main data supporting the results in this study are available within the article and its Supplementary Information. RNA-seq data have been deposited at the NCBI. Gene Expression Omnibus and are accessible under the GEO Series accession number GSE184421.

## Supplementary Information

## Acknowledgements

We thank Danny Labes from the Flow Cytometry facility for his precious technical supervision and support, Luigi Bozzo for his microscopy analysis coaching, Anne-Catherine Clerc from the Cardiovascular Assessment Facility for ultrasonography analysis, Severine Lorrain from the Protein Analysis Facility, Céline Pellaton for the Luminex assays, Jean-Christophe Stehle from the Mouse Pathology facility, the UNIGE IGE3 genomic platform, and the Animal Facilities of UNIL. We are grateful to Alexandre Sarre from the Cardiovascular Assessment Facility for ultrasonography analysis. Special thanks go to Francesco Paolo Ruberto form the Department of Cardiovascular Medicine for his expert help and advice regarding the seahorse experiments. This work was supported by Swiss National Science Foundation grant 310030-169563 to I Stamenkovic, the FNS Sinergia grant CRSII5-177266 to I Stamenkovic, and the Swiss Cancer League grant KLS-4249-08-2017-R to I Stamenkovic. N Riggi is supported by the Swiss National Science Foundation Professorship grant (PP00P3-157468/1 and PP00P3_183724), the Swiss Cancer League grant (KFS-3973-08-2016 and KFS4859-08-2019), the Fond'Action Contre le Cancer Foundation, the FORCE Foundation, and the Fondation Emma Muschamp.

## Author Contributions

S Katakam: conceptualization, data curation, formal analysis, investigation, methodology, and writing—review and editing.
S Anand: formal analysis, investigation, and methodology.
P Martin: investigation.
N Riggi: formal analysis and writing—review and editing.
I Stamenkovic: conceptualization, resources, formal analysis, funding acquisition, and writing—original draft, review, and editing.

## Conflict of Interest Statement

The authors declare that they have no conflict of interest.

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
