## [Reviewer comments · Life Science Alliance]

Life Science Alliance

Necrotic debris and STING exert therapeutically relevant effects on tumor cholesterol homeostasis

Sampath Katakam, Santosh Anand, Patricia Martin, Nicolo Riggi, and Ivan Stamenkovic

DOI: <https://doi.org/10.26508/lsa.202101256>

Corresponding author(s): *Ivan Stamenkovic, University of Lausanne*

Review Timeline:

Submission Date:	2021-10-07
Editorial Decision:	2021-11-02
Revision Received:	2021-12-02
Editorial Decision:	2021-12-08
Revision Received:	2021-12-10
Accepted:	2021-12-10

Transaction Report:

November 2, 2021

Re: Life Science Alliance manuscript #LSA-2021-01256

Prof. Ivan Stamenkovic
University of Lausanne
25 Rue du Bugnon, (CHUV)
Lausanne CH-1011
Switzerland

Dear Dr. Stamenkovic,

Thank you for submitting your manuscript entitled "Necrotic debris and STING exert therapeutically relevant effects on tumor cholesterol homeostasis" to Life Science Alliance. The manuscript was assessed by expert reviewers, whose comments are appended to this letter. We invite you to submit a revised manuscript addressing the Reviewer comments.

Thank you for this interesting contribution to Life Science Alliance. We are looking forward to receiving your revised manuscript.

Sincerely,

B. MANUSCRIPT ORGANIZATION AND FORMATTING:

Reviewer #1 (Comments to the Authors (Required)):

Authors revealed in this study that DAMPs help tumors to grow by promoting cholesterol uptake and synthesis. Moreover, they demonstrated STING signaling increases cholesterol efflux in tumor cells, delaying tumor growth. Therefore, combination therapy using statin and STING agonist would be a rational strategy for patients with tumor. This manuscript is well written and interesting. However, there are some points to be addressed below.

- a) Figure 3D; What cells was used in these data?
- b) Figure 3F; Which is spheroid or adherent?
- c) Figure 4D; Although authors described that DAMPs also induced p65 and TBK phosphorylation, I can't see the band of pTBK1. Please show clear western blotting result and normalized bar graphs.
- d) Also, each lane in western blotting images seems to be like cut and paste... because white gaps are recognized among samples even where to be no band
- e) Figure 4F; Although authors say solvent (CTRL) in its legend, they used C in the figures.
- f) Figure 4G and H; Authors concluded that inhibitory effect of cGAMP treatment on tumors is caused by STING-dependent type I and III interferons. However, it is reported that cGAMP activates STING-mediated cell death pathway in estrogen-receptor negative breast cancer cells (PMID: 33840002). Therefore, only these data sets (Figure 4G and H) cannot eliminate the STING-mediated direct cell death. Does anti-IFNAR1 mAb partially rescue the inhibitory tumor growth by cGAMP treatment?
- g) Authors demonstrated that STING-dependent type I and III IFNs delayed tumor growth in figure 4H. And they also showed cGAMP stimulation causes changes in the expression of genes associated with lipid metabolism in Figure 5. Do the cGAMP-triggered changes also depend on type I and III IFNs?
- h) Figure 6B left panel; Although authors show a significant difference between C and cG in ST cri, I don't think so. Please perform statistical analysis again.
- i) Does cGAMP-dependent cholesterol efflux depend on type I and III IFNs? Please show data demonstrating type I and III IFNs directly increased cholesterol efflux.
- j) Figure 8; Authors can evaluate synergistical effects of cGAMP and statin in vivo setting like figure 7 E, F, and H.

Reviewer #2 (Comments to the Authors (Required)):

Katakam et al aimed to investigate the effect of necrotic debris and damage-associated molecular patterns (DAMPs) on tumor cells. They used Ewing sarcoma cells as their model system. Ewing sarcoma is a pediatric solid tumor with unmet clinical need. Ewing sarcoma tumors display extensive necrosis which is even more pronounced after chemotherapy. The authors argued that the high frequency of relapse and therapy resistance may be linked to the cell debris of necrotic cells.

To test this hypothesis they used Ewing sarcoma cells grown as spheroids and in monolayer in vitro as well as mouse xenotransplant models. Their experimental work shows that DAMPs promote the growth of EwS cells in 3D (spheroids) but not in 2D (adherent) cultures. They also attempted to explain the mechanism(s) driving this observation. To this end they performed gene expression analysis and they found that cholesterol metabolism-associated transcripts may be linked to the tumor growth of 3D cultures. They confirmed this finding by measuring cholesterol uptake after exposure to DAMPs.

Interestingly they found that cholesterol uptake and EwS cell growth (in 3D) is opposed by STING (an endoplasmatic reticulum transmembrane protein that acts as a stimulator of innate immune response) and/or cGAMP stimulation.

Taken together this study touches an unexplored concept in EwS disease progression suggesting that tumor necrosis and cholesterol uptake may be important phenomena that should be further investigated. The opposing effect by STING may suggest novel therapeutic strategies.

Hence, it is important that this study gets published as it provides nice supporting preliminary data in the design of future studies.

Minor points/Suggestions:

- This is a long manuscript overloaded with a lot of experiments (i.e. RT-qPCR) and it is not always easy to follow. The authors may want to consider removing experiments that are not substantial to the main outcome. For instance, it is not clear why they study the impact of genes associated with inflammation (Figure S3A) given that cholesterol metabolism-associated transcripts were enriched only in 3D cultures.
- Figure labelling could be made clearer (i.e. C1, C2 could be replaced by the name of the cell lines).
- Figure S6C - it is not clear why P2/C2 spheres showed opposite response compared to P1/C1 in response to cGAMP. Is there any overlap with data shown in Figure 6B (middle panel)? This part is very confusing. Similarly to the comment above, the authors may want to simplify data presentation.

Reviewer #3 (Comments to the Authors (Required)):

The paper by Katakam et al has assessed the effect of debris released by necrotic cells including damage/danger-associated molecular patterns (DAMPs) on Ewing sarcoma cells grown both in 2D and 3D. The major finding of this paper is that modulation of cholesterol homeostasis is a major effect of necrotic cell debris including activation of the STING pathway. This paper is extremely well written, however there is an overwhelming amount of data provided which at times can make this paper difficult to read. It may be easier to split the data over two publications. The figures are of exceptionally high quality and all experiments are repeated multiple independent times to ensure scientific rigor. All data is well justified.

Minor comments

- P values are not stated within the text, for all conclusions in which the authors state "significance" this should be included. Also please be more descriptive in your results, ie what fold change increase/decrease in expression do you observe.
- Results Section: DAMPs stimulate 3D EWs growth
 - a) Why was camptothecin chosen to induce apoptotic bodies? This agent is not used for the treatment of Ewing sarcoma
 - b) To assess the effect of DAMPs on tumor growth (Figure 1) two different routes of cell administration were used (kidney capsule versus subcutaneous). Comparisons of 2D and 3D growth models can't be compared accurately if the site of colonization is different.
- Results Section: Stimulation of DAMP's causes changes in the expression of genes
 - a) Do EWs cells proliferate faster or slower in 2D or 3D structures? If faster in 3D could this account for the metabolomic differences. Seahorse assays should be conducted (3D and 2D) to assess differences in energy production and metabolism.
- Figure 4D: Densitometry of protein bands is required.
- Figure 4H: Authors state "Both IFN's significantly delayed tumor growth", yet no statistical analysis is shown on the figure.
- Figure S5E is missing from the figure panel.

We would like to thank the reviewers for their thorough and fair review, their encouraging comments and their suggestions, which have helped improve the manuscript. We have taken all of the comments into consideration and our point by point response is detailed below.

Reviewer 1.

a) Figure 3D; What cells was used in these data?

C1 cells were used.

b) Figure 3F; Which is spheroid or adherent?

Spheroids are illustrated in Figure 3F.

c) Figure 4D; Although the authors described that DAMPs also induced p65 and TBK phosphorylation, I can't see the band of pTBK1. Please show clear western blotting results and normalized bar graphs.

We have re-examined the Western blot and corrected an overlooked error: pTBK induced by cGAMP is more pronounced at 1hr than at 3hrs, as indicated in the new Figure 4D. The reviewer is correct in that the pTBK band is not visible in the DAMP lanes. We had stated that « DAMPs also induced p65 and TBK phosphorylation, albeit to a lesser degree than cGAMP (Figure 4D and S4G) » based on the normalized graph but the numbers on the normalized graph for TBK1 are indeed extremely low and do not reflect significant TBK1 phosphorylation following stimulation by DAMPs. This has been rectified in the text. However, there is mild p65 phosphorylation 1 hour following DAMP stimulation, as illustrated by the normalized graph.

d) Also, each lane in western blotting images seems to be like cut and paste... because white gaps are recognized among samples even where to be no band

Different samples were run on different blots. The lanes therefore had to be cut and appropriately associated.

e) Figure 4F; Although authors say solvent (CTRL) in its legend, they used C in the figures.

CTRL has been modified to C to conform to the figures.

f) Figure 4G and H; Authors concluded that inhibitory effect of cGAMP treatment on tumors is caused by STING-dependent type I and III interferons. However, it is reported that cGAMP activates STING-mediated cell death pathway in estrogen-receptor negative breast cancer cells (PMID: 33840002). Therefore, only these data sets (Figure 4G and H) cannot eliminate the STING-mediated direct cell death. Does anti-IFNAR1 mAb partially rescue the inhibitory tumor growth by cGAMP treatment?

We agree with the reviewer and have modified the text, indicating that the IFNs may participate in but are not alone responsible for STING-dependent tumor growth inhibition. Anti-IFN λ antibody does partially rescue the inhibitory effect on tumor growth by cGAMP treatment as illustrated in the figure below. We focused on IFN- λ because it was more strongly induced than IFN- β by cGAMP,

[Figures removed by editorial staff per authors' request].

g) Authors demonstrated that STING-dependent type I and III IFNs delayed tumor growth in figure 4H. And they also showed cGAMP stimulation causes changes in the expression of genes associated with lipid metabolism in Figure 5. Do the cGAMP-triggered changes also depend on type I and III IFNs?

Yes, the changes do partially depend on IFNs. This is illustrated in Figure S6 E and F, which show that IFN β and λ decrease the expression of genes implicated in the mevalonate pathway (E) whereas they augment the expression of genes associated with cholesterol transport (F).

h) Figure 6B left panel; Although authors show a significant difference between C and cG in ST cri, I don't think so. Please perform statistical analysis again.

This was a typo and has been corrected. There is indeed no difference between C and cG in ST cri.

i) Does cGAMP-dependent cholesterol efflux depend on type I and III IFNs? Please show data demonstrating type I and III IFNs directly increased cholesterol efflux

There was no significant difference in cholesterol efflux following exposure of cells to type III IFN, as shown in the figure below. cGAMP-induced cholesterol efflux could be the result of STING signals unrelated to IFN induction.

[Figure removed by editorial staff per authors' request].

j) Figure 8; Authors can evaluate synergistic effects of cGAMP and statin in vivo settings like figure 7 E, F, and H.

We had initiated in vivo experiments upon submitting the manuscript, with the anticipation that this would be a logical request from the reviewers. The experiments are described in the methods section, illustrated in Figure 8F and commented on in the text. In essence, we observed growth inhibitory synergy between cGAMP and statin in tumors with native STING expression. However, in tumor overexpressing STING, this synergy was less obvious, possibly because high expression of STING in conjunction with cGAMP may maximize the effect of altered cholesterol metabolism on tumor growth.

Reviewer 2

- This is a long manuscript overloaded with a lot of experiments (i.e. RT-qPCR) and it is not always easy to follow. The authors may want to consider removing experiments that are not substantial to the main outcome. For instance, it is not clear why they study the impact of genes associated with inflammation (Figure S3A) given that cholesterol metabolism-associated transcripts were enriched only in 3D cultures

Because necrosis triggers an inflammatory response, we felt that it was important to determine whether soluble mediators of inflammation might be responsible for or participate in the increased proliferation of 3D-cultured cells stimulated by DAMPs. Although many cytokines were induced by DAMPs in the tumor cells, they had little or no effect on proliferation in marked contrast to cholesterol. We have confined these results to the supplemental data only.

- Figure labeling could be made clearer (i.e. C1, C2 could be replaced by the name of the cell lines).

We agree with the reviewer, but we opted for the single letter labels because of space limitation on the graphs. The letter code is indicated early on in the text.

Figure S6C - it is not clear why P2/C2 spheres showed opposite response compared to P1/C1 in response to cGAMP. Is there any overlap with data shown in Figure 6B (middle panel)? This part is very confusing. Similarly to the comment above, the authors may want to simplify data presentation. –

We agree with the reviewer. Cholesterol uptake showed variable responses to cGAMP in the different cells used, rendering any conclusion difficult. We have removed these experiments from the revised manuscript.

Reviewer 3

We thank the reviewer for their extremely positive comments. We agree that the manuscript is dense and have considered splitting it into two papers. However, DAMPs and cGAMP are often used for comparison and contrast in several experiments and we concluded that splitting the papers would be difficult.

- P values are not stated within the text, for all conclusions in which the authors state "significance" this should be included. Also please be more descriptive in your results, ie what fold change increase/decrease in expression do you observe.

We have addressed this notion wherever necessary.

- Results Section: DAMPs stimulate 3D EWs growth

a) Why was camptothecin chosen to induce apoptotic bodies? This agent is not used for the treatment of Ewing sarcoma

Camptothecin was used simply as a means to induce apoptosis rather than as a relevant treatment in Ewing sarcoma.

Here are a couple of references as to the use of camptothecin in Ewing sarcoma

- Camptothecin-based regimens for treatment of ewing sarcoma: past studies and future directions. *Sarcoma*. 2011;2011:957957. doi:10.1155/2011/957957
- Preclinical efficacy of the camptothecin-polymer conjugate IT-101 in multiple cancer models

However, camptothecin is indeed not indicated in EwS.

b) To assess the effect of DAMPs on tumor growth (Figure 1) two different routes of cell administration were used (kidney capsule versus subcutaneous). Comparisons of 2D and 3D growth models can't be compared accurately if the site of colonization is different.

We used the kidney capsule for P1 cells only because they, as was the case for several primary tumor-derived cells in our experience, they grow poorly in the subcutaneous compartment. However, all of the experiments using C1 (A673) cells were done by subcutaneous injection (Fig 7).

Results Section: Stimulation of DAMP's causes changes in the expression of genes

a) Do EWs cells proliferate faster or slower in 2D or 3D structures? If faster in 3D could this account for the metabolomic differences. Seahorse assays should be conducted (3D and 2D) to assess differences in energy production and metabolism.

We have conducted seahorse experiments at the reviewer's suggestion and the results are illustrated in Figure 3A. Interestingly, whereas 3D and 2D cultured cell proliferation varied according to the cells used, 3D cultures invariably displayed a far greater spare respiratory capacity than 2D cultures, consistent with the notion that spheroids may be better equipped to withstand various types of stress and possibly also increase their proliferation in response to diverse signals.

- Figure 4D: Densitometry of protein bands is required. -

Densitometry is shown in Figure : S4G

Figure 4H: Authors state "Both IFN's significantly delayed tumor growth", yet no statistical analysis is shown on the figure.

There appeared to be a clear trend, but without statistical significance. This has been rectified in the text.

- Figure S5E is missing from the figure panel.

This was an oversight on our part. The figure has been included.

December 8, 2021

RE: Life Science Alliance Manuscript #LSA-2021-01256R

Prof. Ivan Stamenkovic
University of Lausanne
25 Rue du Bugnon, (CHUV)
Lausanne CH-1011
Switzerland

Dear Dr. Stamenkovic,

Thank you for submitting your revised manuscript entitled "Necrotic debris and STING exert therapeutically relevant effects on tumor cholesterol homeostasis". We would be happy to publish your paper in Life Science Alliance pending final revisions necessary to meet our formatting guidelines.

- please add an Abstract to our system
- please make sure the author order in your manuscript and our system match
- there is a callout for figure S6G in the manuscript text, but no such panel in the figure. Please revise
- please add callouts for Figures S3B, S6C, and S7B-D to your main manuscript text.

FIGURE CHECKS:

- please add molecular weights next to all blots
- the scale bar for Figure 5G is hard to see

A. FINAL FILES:

B. MANUSCRIPT ORGANIZATION AND FORMATTING:

Sincerely,

December 10, 2021

RE: Life Science Alliance Manuscript #LSA-2021-01256RR

Prof. Ivan Stamenkovic
University of Lausanne
25 Rue du Bugnon, (CHUV)
Lausanne CH-1011
Switzerland

Dear Dr. Stamenkovic,

Thank you for submitting your Research Article entitled "Necrotic debris and STING exert therapeutically relevant effects on tumor cholesterol homeostasis". It is a pleasure to let you know that your manuscript is now accepted for publication in Life Science Alliance. Congratulations on this interesting work.

DISTRIBUTION OF MATERIALS:

Again, congratulations on a very nice paper. I hope you found the review process to be constructive and are pleased with how the manuscript was handled editorially.

Sincerely,
